PREPARED FOR SUBMISSION TO JHEP

# Resurgence and renormalons in the one-dimensional Hubbard model

**Marcos Mariño and Tomás Reis**

*Département de Physique Théorique et Section de Mathématiques*
*Université de Genève, Genève, CH-1211 Switzerland*

*E-mail:* Marcos.Marino@unige.ch, Tomas.Reis@unige.ch

ABSTRACT: We use resurgent analysis to study non-perturbative aspects of the one-dimensional, multicomponent Hubbard model with an attractive interaction and arbitrary filling. In the two-component case, we show that the leading Borel singularity of the perturbative series for the ground-state energy is determined by the energy gap, as expected for superconducting systems. This singularity turns out to be of the renormalon type, and we identify a class of diagrams leading to the correct factorial growth. As a consequence of our analysis, we propose an explicit expression for the energy gap at weak coupling in the multi-component Hubbard model, at next-to-leading order in the coupling constant. In the two-component, half-filled case, we use the Bethe ansatz solution to determine the full trans-series for the ground state energy, and the exact form of its Stokes discontinuity.

## 1  Introduction

Many important phenomena in quantum physics can not be captured by perturbation theory at weak coupling. These include tunneling in quantum mechanics, condensates in asymptotically free theories, or energy gaps in superconducting systems. All these effects are exponentially small in the coupling constant, and therefore they are invisible in the standard perturbative approach. Incorporating these effects in a systematic way remains a challenge for theoretical physics. A possible framework to do that is the theory of resurgence, which grew up from the efforts of physicists and mathematicians to understand perturbation theory at large orders and its connection to non-perturbative effects. In the theory of resurgence, conventional perturbative series are extended to *trans-series*, which include exponentially small effects explicitly. The perturbative and non-perturbative sectors of these trans-series turn out to be linked in a precise way (hence the name "resurgence"), and physical observables are expected to be obtained by generalized Borel resummation. In recent years, the theory of resurgence has been applied to many different problems, from quantum mechanics to string theory, see e.g. [1–3] for reviews and references.

In [4, 5] we used the theory of resurgence to study quantum many-body systems. Specifically, we focused on Fermi systems with attractive interactions, which are known to develop a gap in

the spectrum. This gap is non-perturbative in the coupling constant, as one can show e.g. in BCS theory. In [5], based on the ideas of resurgence, we conjectured that this gap determines the large order behavior of conventional many-body perturbation theory. We gave substantial evidence for this conjecture in an integrable one-dimensional model, the Gaudin–Yang model [6, 7], and we showed that the factorial growth of perturbation theory is of the renormalon type, i.e. it is due to special types of diagrams, after integration over the momenta. We concluded that the ground-state energy of the Gaudin–Yang model must be given by a trans-series, encoding non-perturbative effects due to these renormalons. Partial evidence for this conjecture was also given for other models.

In this paper we continue this line of research and we consider another important model for interacting fermions, namely the multi-component, attractive Hubbard model in one-dimension. The Hubbard model is a paradigmatic example of a strongly interacting many-body system and it has been extensively studied. In one dimension, the two-component case can be solved with the Bethe ansatz [8]. This has made it possible to combine integrability techniques with more conventional approaches in many-body theory in order to understand the physics of strongly correlated electrons (see e.g. [9]). For these reasons, the Hubbard model is an ideal laboratory to test the conjecture of [5], and more generally, the ideas of resurgence, in a more complicated setting. Indeed, the one-dimensional Hubbard model at arbitrary filling can be regarded as a deformation of the Gaudin–Yang model studied in [5], and the latter can be recovered from the former in a double-scaling limit of small density and weak coupling.

In the case of the two-component Hubbard model, we use integrability to study in detail the perturbative expansion of the ground state energy, and we verify the conjecture of [5] connecting the large order behavior of this expansion to the energy gap. In addition, we show that the corresponding Borel singularity can be explained by the renormalon behavior of ring diagrams. As a consequence of this analysis, we are able to extract the form of the energy gap in the multi-component case, at weak coupling (and under some mild assumptions). This is a prediction of our resurgent analysis for a non-integrable model.

As we have just recalled, perturbative series should be extended to trans-series in order to incorporate non-perturbative effects. The calculation of perturbative series is an essential tool in quantum theory which is relatively well-understood, at least in principle. The calculation of trans-series is a different matter. When they are associated to instantons, trans-series can be calculated by expanding the path integral around a non-trivial saddle point. When trans-series are related to renormalon effects, there is no first-principle method to obtain them from the path integral. In some cases they can be calculated by using the operator product expansion (OPE), but when this is not possible, one has to rely on the large order analysis of the perturbative series. This has been exploited in the study of non-perturbative corrections to certain observables in QCD, like the pole mass of a quark (see e.g. [10] and references therein). In practice, one looks at particular families of renormalon diagrams and tries to extract some non-perturbative information from them.

In this paper we will adopt this point of view and we will study the trans-series associated to the ground state energy density of the Hubbard model by looking at the large order behavior of perturbation theory. By focusing on ring diagrams, we will extract an approximate trans-series, similar to what is done in QCD in the so-called large $\beta_0$ approximation. However, it turns out that, at half-filling, one can do better, and we can improve substantially the results obtained in [5] for the Gaudin–Yang model. The reason is that, in that case, the perturbative series for the ground state energy density can be written down explicitly, as first found by Misurkin and Ovchinnikov in [11]. This is, to our knowledge, one of the few perturbative series in quantum

theory where the coefficients are known in closed form to all orders. There are other examples where this happens, like Chern–Simons theory on certain three-manifolds and some simplified models of string theory. However, the one-dimensional Hubbard model at half-filling is unique in the sense that its non-perturbative effects are due to renormalons, and not to instantons. Building on the result of [11], it is possible to determine the full, explicit trans-series for the ground state energy energy, including all non-perturbative corrections due to renormalons. This leads to an exact expression for its Stokes discontinuity, which plays an important rôle in the mathematical theory of resurgence.

The paper is organized as follows. In section 2 we review the Hubbard model and some of the techniques we will use in this paper to study it. Although many of the results we present are well-known, others seem to be new (like the explicit expressions for ring diagrams or the BCS analysis at arbitrary filling). In section 3 we study the large order behavior of the perturbative series at arbitrary filling, and we show that it is not Borel summable. Its factorial growth is controlled by the energy gap, in accord with the conjecture of [5]. We also show that the corresponding Borel singularity is of the renormalon type, and we show that ring diagrams lead to the correct singularity, similarly to what was found in [5] in the Gaudin–Yang model. By putting together various ingredients, we propose an explicit formula for the weak-coupling behavior of the energy gap in the multi-component case. In section 4 we use more advanced tools of resurgent analysis. We focus mostly on the half-filled case and we determine the full trans-series expansion of the ground state energy. The last section collects our conclusions and prospects for future work. Appendix A explains how to obtain analytically the coefficients of the perturbative series of the ground state energy, as power series in $n$. Appendix B gives another derivation of the trans-series in the model at half-filling, based on the original integral representation of [11].

## 2 The one-dimensional Hubbard model

### 2.1 The perturbative approach

Our focus in this paper will be on the multi-component Hubbard model in one dimension, see for example [9] for a comprehensive reference. In this model we have $N$ fermions in a one-dimensional lattice with $L$ sites. Each fermion comes in $\kappa$ flavors, and the model has an internal $U(\kappa)$ global symmetry. The case $\kappa = 2$ corresponds to the standard spin $1/2$ case. The creation/annihilation operator for a fermion at the site $i$ of the lattice, and with flavor index $\sigma$ will be denoted by $c_{i\sigma}^\dagger$ (respectively, $c_{i\sigma}$). The Hamiltonian of the one-dimensional Hubbard model has the form

$$\mathsf{H} = \mathsf{H}_0 + \mathsf{H}_I. \tag{2.1}$$

Here, $\mathsf{H}_0$ is the kinetic or hopping energy

$$\mathsf{H}_0 = \sum_{i,j,\sigma} t_{ij} c_{i\sigma}^\dagger c_{j\sigma} = \sum_{k,\sigma} \epsilon_k a_{k\sigma}^\dagger a_{k\sigma}. \tag{2.2}$$

We have denoted the Fourier transforms of $c_{i\sigma}^\dagger$, $c_{i\sigma}$ to momentum space by $a_{k\sigma}^\dagger$, $a_{k\sigma}$. We will consider the standard next-neighbor interaction, which gives

$$\epsilon_k = -t\cos(k), \tag{2.3}$$

and we will set $t = 1$. The interacting part of the Hamiltonian, $\mathsf{H}_I$, is given by

$$\mathsf{H}_I = -u \sum_j \sum_{\sigma,\tau} n_{j\sigma} n_{j\tau} \tag{2.4}$$

where $u$ is the coupling constant and

$$n_{j\sigma} = c_{j\sigma}^\dagger c_{j\sigma} \tag{2.5}$$

is the number operator. The interaction can be written in momentum space as

$$\mathsf{H}_I = -\frac{u}{L} \sum_{k,k',q} \sum_{\sigma,\tau} a_{k+q,\sigma}^\dagger a_{k,\sigma} a_{k'-q,\tau}^\dagger a_{k',\tau}. \tag{2.6}$$

We will assume that $u > 0$, which corresponds, with our sign conventions, to an attractive interaction. The density or filling of the model is defined by

$$n = \frac{N}{L} \tag{2.7}$$

and is a real number $0 \le n \le 1$. In the two-component case, the value $n = 1$ is usually referred to as half-filling.

In this paper we will focus on the ground state energy density of this model in the thermodynamic limit in which $N, L \to \infty$ while the density $n$ is fixed. This ground state energy density is a function of $u$, $\kappa$ and $n$, and we will denote it by $E(u, n; \kappa)$. The first approach to computing this quantity is perturbation theory at small $u$, which is an expansion around the non-interacting theory with $u = 0$. The Fermi momentum is given by

$$k_F = \frac{\pi \widetilde{n}}{2}, \qquad \widetilde{n} = \frac{2n}{\kappa}. \tag{2.8}$$

Our choice of variables is such that, in the two-component case, $\widetilde{n} = n$.

There are various interesting regimes in which we can study the ground state energy, as we will show in more detail in this section:

1. The limit $u \to 0$ with $n, \kappa$ fixed corresponds to the weak coupling, perturbative expansion.

2. We can consider a double-scaling limit in which

$$u \to 0, \qquad n \to 0, \tag{2.9}$$

in such a way that $\kappa$ and

$$\frac{n}{u} = \frac{1}{\gamma} \tag{2.10}$$

are fixed. In this limit one recovers the multi-component Gaudin–Yang model [6, 7], describing a one-dimensional, non-relativistic gas of fermions with $\kappa$ components and interacting through a delta function potential.

3. One can consider another double-scaling limit, in which

$$u \to 0, \qquad \kappa \to \infty, \tag{2.11}$$

in such a way that

$$\upsilon = \kappa u \tag{2.12}$$

and the Fermi momentum (2.8), or equivalently $\tilde{n}$, are fixed. Note in particular that, to keep $\tilde{n}$ fixed, one has to take $n \to \infty$ in this limit. This is the large $\kappa$ or 't Hooft limit.

Let us first consider the standard weak-coupling expansion. In the absence of interaction, the ground state energy density is given by

$$E(u = 0, n; \kappa) = -\frac{2\kappa}{\pi} \sin\left(\frac{\pi \widetilde{n}}{2}\right). \tag{2.13}$$

The corrections to the free theory can be computed in standard stationary perturbation theory. This leads to an asymptotic, formal power series in $u$,

$$E(u, n; \kappa) = -\frac{2\kappa}{\pi} \sin\left(\frac{\pi \widetilde{n}}{2}\right) + E_{\mathrm{HF}}(u, n; \kappa) + \sum_{\ell \geq 2} E_\ell(n; \kappa) u^\ell. \tag{2.14}$$

In the expression in the r.h.s., the second term is the Hartree–Fock correction to the energy, which is given by

$$E_{\mathrm{HF}}(u, n; \kappa) = -\frac{1}{4}\kappa(\kappa - 1)u\widetilde{n}^2. \tag{2.15}$$

When no reference to $\kappa$ is explicitly indicated in an expression, the value $\kappa = 2$ should be implicitly understood. For example, we will denote

$$E_\ell(n) := E_\ell(n; \kappa = 2). \tag{2.16}$$

Let us now consider the double-scaling limit (2.9), where one recovers the multicomponent Gaudin–Yang model. This model is characterized by a density $n_{\mathrm{GY}}$, a dimensionless coupling $\gamma$, and the number of flavors $\kappa$. It is useful to introduce the following quantities [5]:

$$\lambda = \left(\frac{\kappa}{2}\right)^2 \gamma, \qquad e_{\mathrm{GY}}(\lambda; \kappa) = \frac{1}{4}\frac{E_{\mathrm{GY}}/\kappa}{(n_{\mathrm{GY}}/\kappa)^3} = \sum_{\ell \geq 0} e_\ell(\kappa)\lambda^\ell, \tag{2.17}$$

where $E_{\mathrm{GY}}$ is the ground state energy density of the Gaudin–Yang model. In the double-scaling limit described by (2.9) and (2.10), the coupling $\gamma$ is obtained from (2.10), and the ground state energy density of the Hubbard model has the expansion,

$$E(u, n; \kappa) = -\kappa\widetilde{n} + \widetilde{n}^3 e_{\mathrm{GY}}(\lambda; \kappa) + \mathcal{O}(\widetilde{n}^4). \tag{2.18}$$

In particular, one has the following expansion of the coefficients $E_\ell(n; \kappa)$ as $n \to 0$,

$$E_\ell(n; \kappa) = \widetilde{n}^{3-\ell} \sum_{r \geq 0} E_\ell^{(r)}(\kappa)\widetilde{n}^{2r}, \tag{2.19}$$

where the leading coefficient as $\widetilde{n} \to 0$ agrees with the coefficient in the perturbative expansion (2.17) of the Gaudin–Yang model:

$$E_\ell^{(0)}(\kappa) = e_\ell(\kappa). \tag{2.20}$$

Calculating the series (2.14) by using conventional perturbation theory is in general difficult, since the number of diagrams grows factorially with the order. A detailed analysis of the very first terms was performed in [12]. It is then interesting to consider the double-scaling or large $\kappa$ limit. In this limit, the ground state energy is dominated by ring diagrams (see e.g. [13, 14]). More precisely, the ground state energy has an expansion in powers of $1/\kappa$ given by

$$\frac{1}{\kappa} E(u, n; \kappa) = \sum_{r=0}^{\infty} e_r(\upsilon, \widetilde{n})\kappa^{-r}. \tag{2.21}$$

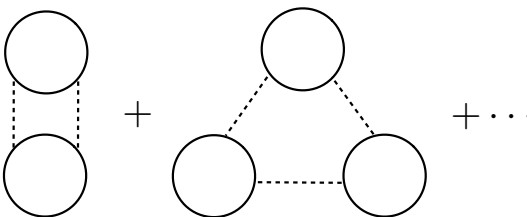

**Figure 1**. Ring diagrams with $\ell \geq 2$ bubbles.

The leading contribution comes from the free theory and the Hartree diagram,

$$e_0(v, \widetilde{n}) = -\frac{2}{\pi} \sin\left(\frac{\pi \widetilde{n}}{2}\right) - \frac{1}{4} v \widetilde{n}^2, \tag{2.22}$$

while the subleading term $e_1(v, \widetilde{n})$ is a sum over the Fock diagram and all ring diagrams, represented in Fig. 1. This sum will play an important rôle in this paper, so let us calculate it in some detail. The approximation in which we keep $e_0$ and $e_1$ in (2.21) is sometimes called the random phase approximation or RPA.

The contribution of a ring diagram with $\ell \geq 2$ bubbles to the ground state energy density is given by (see e.g. [14], section 7.7.4)

$$E_\ell^{\mathrm{ring}}(n; \kappa) = -\frac{(-2\kappa)^\ell}{\ell} \int_{-\pi}^{\pi} \frac{\mathrm{d}q}{2\pi} \int_0^\infty \frac{\mathrm{d}\omega}{2\pi} \left(\Pi(q, \mathrm{i}\omega)\right)^\ell, \tag{2.23}$$

where $\Pi(q, \mathrm{i}\omega)$ is the polarization function with imaginary frequency. It can be obtained in closed form by direct integration. Let us introduce the following functions:

$$S(\omega) = \frac{2}{\sqrt{1 + \omega^2}}, \qquad P(q, \omega) = \frac{1 + \omega^2 \tan^2(q/4)}{1 + \omega^2}, \tag{2.24}$$

as well as

$$F(q, \omega) = -\frac{1}{\sqrt{1 + \omega^2}} \log\left\{\frac{1 + S(\omega) + P(q, \omega)}{1 - S(\omega) + P(q, \omega)}\right\}. \tag{2.25}$$

Then,

$$\Pi(q, \mathrm{i}\omega) = -\frac{1}{8\pi \sin\left(\frac{q}{2}\right)} \left\{F\left(q + \pi\widetilde{n}, \frac{\omega}{4 \sin\left(\frac{q}{2}\right)}\right) - F\left(q - \pi\widetilde{n}, \frac{\omega}{4 \sin\left(\frac{q}{2}\right)}\right)\right\}. \tag{2.26}$$

The subleading contribution to the ground state energy in the large $\kappa$ limit is then given by

$$e_1(v, \widetilde{n}) = \frac{v\widetilde{n}^2}{4} - \sum_{\ell \geq 2} \frac{(-2v)^\ell}{\ell} \int_{-\pi}^{\pi} \frac{\mathrm{d}q}{2\pi} \int_0^\infty \frac{\mathrm{d}\omega}{2\pi} \left(\Pi(q, \mathrm{i}\omega)\right)^\ell. \tag{2.27}$$

As an application of this result, we note that the second order coefficient $E_2(n; \kappa)$ in (2.14) is given solely by the ring diagram contribution $E_2^{\mathrm{ring}}(n; \kappa)$, albeit with the spin factor $\kappa(\kappa - 1)$ instead of $\kappa^2$. Therefore, we find

$$E_2(n; \kappa) = -2\kappa(\kappa - 1) \int_{-\pi}^{\pi} \frac{\mathrm{d}q}{2\pi} \int_0^\infty \frac{\mathrm{d}\omega}{2\pi} \Pi^2(q, \mathrm{i}\omega). \tag{2.28}$$

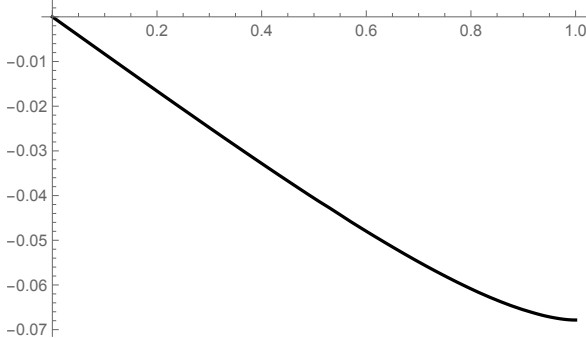

**Figure 2**. The coefficient $E_2(n)$ in (2.14) as a function of $0 \le n \le 1$.

In Fig. 2 we have plotted $E_2(n; \kappa)$, as a function of $n$, for $\kappa = 2$. It agrees with the result in [12]. Note in addition that

$$E_2(n) \sim -\frac{n}{12}, \qquad n \to 0, \tag{2.29}$$

in agreement with (2.20) and the perturbative result for the Gaudin–Yang model [4, 5, 15].

As we will review in section 2.3, it is possible to use the integrability of the two-component Hubbard model to obtain further information on the functions $E_\ell(n)$. Based on the result of Lieb and Wu [8], Misurkin and Ovchinnikov found in [11] an explicit formula for these coefficients in the half-filled case, i.e. when $n = 1$[1]. Their result reads:

$$E_{2k}(n=1) := h_k = -\frac{(2k-1)(2^{2k+1}-1)((2k-3)!!)^3}{2^{5k-3}(k-1)!} \frac{\zeta(2k+1)}{\pi^{2k+1}}, \qquad k \ge 1, \tag{2.30}$$

$$E_{2k+1}(n=1) = 0.$$

When $n < 1$, obtaining the coefficients $E_\ell(n)$ explicitly as a function of $n$ from the Bethe ansatz turns out to be technically difficult. As we will show in section 2.3 and Appendix A, one can obtain them however as a power series in $n$ around $n = 0$, as in (2.19).

## 2.2 The BCS approach

In the two-component case, the ground state of the attractive, one-dimensional Hubbard model is made out of Cooper pairs and can be regarded as a superconductor. It is then natural to use BCS theory to describe the model, and in particular to obtain an approximate value for the energy gap. This was done in [18], although the BCS gap was only computed there in the case of half-filling $n = 1$. We will extend the calculation of [18] to obtain the BCS gap for arbitrary $n \in (0, 1]$.

The general BCS equations for the Hubbard model in arbitrary dimension can be found in [19]. We introduce the shifted chemical potential which takes into account the Hartree correction:

$$\hat{\mu} = \mu + un. \tag{2.31}$$

We also introduce the function

$$E_{\mathbf{k}} = \sqrt{(\epsilon_{\mathbf{k}} - \hat{\mu})^2 + \Delta_{\text{BCS}}^2}, \tag{2.32}$$

---

[1]In the capitalist bloc, this result was rederived seven years later [16], building on results of [17].

where $\Delta_{\mathrm{BCS}}$ is the BCS gap. There are two BCS equations. The first one is the gap equation,

$$\frac{1}{u} = \frac{1}{L} \sum_{\mathbf{k}} \frac{1}{E_{\mathbf{k}}} \tag{2.33}$$

while the second equation determines the density,

$$n = \frac{1}{L} \sum_{\mathbf{k}} \left( 1 - \frac{\epsilon_{\mathbf{k}} - \hat{\mu}}{E_{\mathbf{k}}} \right). \tag{2.34}$$

Solving the two equations simultaneously one finds the gap and the chemical potential $\hat{\mu}$ as functions of the coupling constant $u$ and the filling $n$.

In the one-dimensional case, where $\epsilon_k$ is given by (2.3), we find

$$\frac{2\pi}{u} = \int_{-\pi}^{\pi} \frac{\mathrm{d}k}{\sqrt{(2\cos k + \hat{\mu})^2 + \Delta_{\mathrm{BCS}}^2}},$$
$$n = 1 + \frac{1}{2\pi} \int_{-\pi}^{\pi} \frac{2\cos k + \hat{\mu}}{\sqrt{(2\cos k + \hat{\mu})^2 + \Delta_{\mathrm{BCS}}^2}} \mathrm{d}k. \tag{2.35}$$

The integrals appearing in (2.35) are elliptic, and they can be calculated explicitly with formulae in e.g. [20]. If we introduce

$$\rho_{\pm} = \sqrt{\frac{(\Delta_{\mathrm{BCS}} \pm 2\mathrm{i})^2 + \hat{\mu}^2}{\Delta_{\mathrm{BCS}}^2 + (\hat{\mu} - 2)^2}}, \qquad m = -\frac{\rho_+ - \rho_-}{4\rho_+\rho_-}, \qquad \alpha = \Delta_{\mathrm{BCS}}^2 + (\hat{\mu} - 2)^2, \tag{2.36}$$

we find that the gap equation can be written as

$$\frac{2\pi}{u} = \frac{4K(m)}{\sqrt{\alpha\rho_+\rho_-}}, \tag{2.37}$$

where $K(m)$ is the elliptic integral of the first kind. The equation for the filling is

$$n = 1 + \frac{1}{2\pi} \frac{4}{\sqrt{\alpha\rho_+\rho_-}} \left\{ \left( \frac{\alpha}{2\hat{\mu}} (\rho_+\rho_- + 1) + 2 - \hat{\mu} \right) K(m) - 2\frac{\rho_+\rho_- + 1}{\rho_+\rho_- - 1} \Pi(\tilde{m}, m) \right\}, \tag{2.38}$$

where

$$\tilde{m} = \frac{1}{2} \left( 1 - \frac{4 + \Delta_{\mathrm{BCS}}^2 + 4}{\alpha\rho_+\rho_-} \right) \tag{2.39}$$

and $\Pi(\tilde{m}, m)$ is the elliptic integral of the third kind.

The above equations are exact. We are interested in deriving an expression for $\Delta_{\mathrm{BCS}}$ at weak coupling but arbitrary filling. This means expanding the elliptic integral of the first kind around $m = 1$. From the gap equation we find

$$\Delta_{\mathrm{BCS}} \approx 2(4 - \hat{\mu}^2) \exp\left( -\frac{\pi}{u} \sqrt{1 - \frac{\hat{\mu}^2}{4}} \right). \tag{2.40}$$

By plugging this expansion into the equation for the filling, we obtain

$$\hat{\mu} = \mu_0 + \mu_1 \Delta_{\mathrm{BCS}}^2 + \mathcal{O}(\Delta_{\mathrm{BCS}}^4), \tag{2.41}$$

where

$$\mu_0 = 2\cos\left(\frac{\pi n}{2}\right), \quad \mu_1 = \frac{3\cos\left(\frac{\pi n}{2}\right) + \frac{\pi}{u}\sin(\pi n)}{4(1 - \cos(\pi n))}. \tag{2.42}$$

We conclude that

$$\Delta_{\text{BCS}} \approx 8\sin^2\left(\frac{\pi n}{2}\right)\exp\left(-\frac{\pi}{u}\sin\left(\frac{\pi n}{2}\right)\right), \tag{2.43}$$

which is the approximate expression for the gap obtained in BCS theory. We note that this is exponentially small in the coupling constant $u$, so it is a non-perturbative effect. As we will see, the BCS approximation captures correctly the exponential part of the gap, but not the prefactor. We will show in this paper, following the ideas of [5], that the exponent appearing in (2.43) is related to the factorial divergence of the perturbative series.

## 2.3 The Bethe ansatz approach

In the two-component case, the ground state energy of the Hubbard model can be obtained exactly by using the Bethe ansatz, as first noted by Lieb and Wu [8] (see [21] for a pedagogical introduction to the Bethe ansatz solution, and [9] for a comprehensive review). In this paper we are interested in the attractive regime with arbitrary filling and zero magnetization. In the thermodynamic limit, the properties of the ground state are encoded in a single function $f(x)$, describing the distribution of Bethe roots. It satisfies the equation

$$\frac{f(x)}{2} + \frac{1}{2\pi}\int_{-B}^{B}\frac{f(x')}{1 + (x - x')^2} = \text{Re}\frac{1}{\sqrt{1 - (x - \mathrm{i}/2)^2 u^2}}. \tag{2.44}$$

The endpoint of the interval $[-B, B]$ where $f(x)$ is defined is determined implicitly by the filling, through the equation

$$\frac{n}{u} = \frac{1}{\pi}\int_{-B}^{B}f(x)\mathrm{d}x, \tag{2.45}$$

and the ground state energy density is then given by

$$E(u, n) = -\frac{2u}{\pi}\int_{-B}^{B}\text{Re}\sqrt{1 - (x - \mathrm{i}/2)^2 u^2}\, f(x)\mathrm{d}x. \tag{2.46}$$

It is easy to verify that, in the double-scaling limit (2.9)-(2.10), the integral equation (2.44) reduces to the Gaudin integral equation governing the ground state of the Gaudin–Yang model. In particular, $f(x)$ becomes the Gauding–Yang distribution.

At half-filling $n = 1$, one has $B \to \infty$, and the integral equation (2.44) can be solved exactly by Fourier transformation. The distribution of Bethe roots is

$$f(x) = \int_{\mathbb{R}}\frac{\mathrm{e}^{-\mathrm{i}ukx}J_0(k)}{2\cosh(uk/2)}\mathrm{d}k, \tag{2.47}$$

and the ground state energy density is given by [8, 22]

$$E(u, n = 1) = -u - 4\mathcal{I}(u), \tag{2.48}$$

where

$$\mathcal{I}(u) = \int_0^\infty \frac{\mathrm{d}\omega}{\omega}\frac{J_0(\omega)J_1(\omega)}{1 + \mathrm{e}^{u\omega}}. \tag{2.49}$$

Here, $J_{0,1}(\omega)$ are Bessel functions. The asymptotic expansion of this function around $u = 0$ was worked out in [11], and one obtains in this way the result (2.30) for the coefficients of the perturbative expansion.

In addition to the ground state energy, the Bethe ansatz makes it possible to calculate the energy gap between the ground state and the first excited state. The weak-coupling expansion of the gap, at next-to-leading order, was computed in [23, 24] for arbitrary filling, and one finds

$$\Delta \approx \frac{4}{\pi} \sqrt{2 \sin^3 \left( \frac{\pi n}{2} \right)} u^{1/2} \exp \left( -\frac{\pi}{u} \sin \left( \frac{\pi n}{2} \right) \right), \qquad u \to 0. \tag{2.50}$$

As noted before, the leading, exponentially small dependence on the coupling (i.e. the coefficient in the exponent) agrees with the BCS calculation (2.43). However, the sub-leading dependence (i.e. the dependence on the coupling constant in the prefactor of the exponential) is not captured correctly by the BCS approximation. At half-filling it is possible to obtain an all-orders expression for the gap, given by [8, 18]

$$\Delta = \frac{8}{\pi} \sum_{r=0}^{\infty} \frac{1}{2r+1} K_1 \left( \frac{\pi}{u} (2r+1) \right), \tag{2.51}$$

where $K_1(z)$ is a modified Bessel function. At weak coupling, this is an infinite sum of exponentially small effects of the form $\mathrm{e}^{-(2r+1)\pi/u}$. The first exponential, corresponding to $r = 0$, agrees with (2.50) when $n = 1$.

## 3  Large order behavior and the energy gap

### 3.1  General aspects

One of the main aspects of the theory of resurgence (in fact, the aspect which is responsible for its name) is that non-perturbative effects appear or "resurge" in the large order behavior of the perturbative series. Following this logic, we proposed in [5] that, in a weakly interacting Fermi system with an attractive interaction, the perturbative series is not Borel summable and that the first singularity in the Borel plane is determined by the energy gap. Let us now briefly review the relationship between large order behavior, non-perturbative effects and Borel singularities. More sophisticated tools of the theory of resurgence will be deployed in section 4.

Let

$$\varphi(z) = \sum_{k \geq 0} a_k z^k \tag{3.1}$$

be a factorially divergent formal power series. More precisely, let us assume that the coefficients $a_k$ grow with $k$ as

$$a_k \sim \frac{\mu_0}{2\pi} A^{-k-b} \Gamma(k+b), \qquad k \gg 1. \tag{3.2}$$

It is easy to show (see e.g. [25]) that this growth leads to an exponentially small non-perturbative effect of the form

$$\mu_0 z^{-b} \mathrm{e}^{-A/z}, \qquad z \to 0. \tag{3.3}$$

As it will be clarified in section 4, the proper framework to understand these effects is the theory of trans-series. The Borel transform of $\varphi(z)$ is defined as

$$\widehat{\varphi}(\zeta) = \sum_{k \geq 0} \frac{a_k}{k!} \zeta^k, \tag{3.4}$$

and it follows from (3.2) that it has a singularity at $\zeta = A$.

According to the conjecture in [5], the large order behavior of the series giving the ground state energy (2.14) should be closely related to the square of the energy gap. More precisely, the parameters $A(n; \kappa)$ and $b(n; \kappa)$ controlling the large order behavior of (2.14)

$$E_\ell(n; \kappa) \sim A(n; \kappa)^{-\ell - b(n;\kappa)} \Gamma\left(\ell + b(n; \kappa)\right), \qquad \ell \gg 1, \tag{3.5}$$

should be related to the weak-coupling behavior of the energy gap

$$\Delta \approx u^{-b(n;\kappa)/2} \exp\left(-\frac{A(n; \kappa)}{2u}\right), \qquad u \to 0. \tag{3.6}$$

This conjecture was verified in [5], when $\kappa = 2$, in the cases $n = 1$ and $n \to 0$. In the half-filled case, the perturbative series is given explicitly in (2.30) and the asymptotics can be actually derived analytically. We will deepen this analysis in section 4. The case $n \to 0$ corresponds to the Gaudin–Yang model, where the conjecture was tested by studying numerically the first 50 terms of the perturbative series. In this section we will provide detailed evidence that the conjecture is true for arbitrary $n$, in the two-component case.

### 3.2   The two-component case

In the case of $\kappa = 2$, by comparing the general expression for the gap in (2.50) with (3.6), we obtain

$$A(n) = 2\pi \sin\left(\frac{\pi n}{2}\right), \qquad b(n) = -1. \tag{3.7}$$

According to the general conjecture in [5], the large order behavior of the series (2.14) should be given by

$$E_\ell(n) \sim c(n) \left(2\pi \sin\left(\frac{\pi n}{2}\right)\right)^{-\ell + 1} \Gamma(\ell - 1), \qquad \ell \gg 1. \tag{3.8}$$

As a direct test of this conjecture, we could produce a large number of coefficients in the series, and check the asymptotic behavior (3.8). The calculation of the coefficients $E_\ell(n)$ directly in perturbation theory is not feasible, so one could try to use the Bethe ansatz equations. It is known that the weak-coupling limit of these equations is difficult to analyze. However, a new technique due to Volin [26, 27] made it possible to solve this problem in the Gaudin–Yang model [4, 5] and in other one-dimensional models, like the Lieb–Liniger model [4]. Volin's technique was originally developed in the context of relativistic, integrable field theories in two dimensions, and has been further developed in [28–30]. It can be also applied to the problem of the capacitance of a circular plate capacitor [4, 31].

It turns out that, due to the form of the r.h.s. of the integral equation (2.44), it is difficult to extend Volin's technique to this case for arbitrary $n$, and to obtain the form of the coefficients $E_\ell(n)$ in closed form[2]. Still, it is possible to find a systematic solution by expanding around the Gaudin–Yang limit $n = 0$. This provides a solution for the distribution $f(x)$ and the coefficients $E_\ell(n)$ as power series in $n^2$. The method is explained in Appendix A. One obtains an expansion of the form (see (2.19))

$$\frac{E_\ell(n)}{n^{3-\ell}} = \sum_{k=0}^{\infty} E_\ell^{(k)} n^{2k}. \tag{3.9}$$

---

[2]This is in contrast to the strong coupling expansion at large $u$, where the corresponding coefficients can be obtained as functions of $n$ in closed form [32].

As a simple example, we find for the function $E_2(n)$ plotted in Fig. 2 the following expansion:

$$E_2(n) = -\frac{n}{12} + \frac{\left(12 - \pi^2\right)n^3}{288} + \frac{\left(12\pi^2 - \pi^4\right)n^5}{4608} + \left(\frac{\pi^4}{5760} - \frac{61\pi^6}{3870720}\right)n^7$$
$$+ \left(\frac{121\pi^6}{9289728} - \frac{277\pi^8}{222953472}\right)n^9 + \left(\frac{3917\pi^8}{3715891200} - \frac{50521\pi^{10}}{490497638400}\right)n^{11} + \mathcal{O}(n^{13}).$$

(3.10)

Since $n$ is a parameter, we can find a prediction for the behavior of the sequence $E_\ell^{(k)}$ for fixed $k$ and large $\ell$ by simply expanding (3.8). The prefactor $c(n)$ appearing in (3.8) satisfies

$$c(n) = -\frac{n^2}{\pi}\left(1 + \mathcal{O}(n^2)\right), \qquad n \to 0.$$

(3.11)

This follows from the Gaudin–Yang limiting behavior as $n \to 0$, obtained in [5]. Let us now introduce the coefficients $s_k(\ell)$ by

$$\left(\frac{t}{\sin(t)}\right)^{\ell-1} = \sum_{k=0}^{\infty} s_k(\ell)t^{2k},$$

(3.12)

which are polynomials in $\ell$ of degree $k$. Let us denote by $\sigma_k$ the coefficient of the highest power of $\ell$ in $s_k(\ell)$:

$$s_k(\ell) = \sigma_k \ell^k + \cdots$$

(3.13)

It is then easy to show that the behavior of the sequence $E_\ell^{(k)}$ at fixed $k$ and large $\ell$ is given by

$$E_\ell^{(k)} \sim -\sigma_k \frac{\pi^{2k+1}}{4^k}\pi^{-2\ell}\Gamma(\ell + k - 1), \qquad \ell \gg 1.$$

(3.14)

We find, for example, for $k = 1$ and $k = 2$,

$$E_\ell^{(1)} \sim -\frac{\pi^3}{24}\pi^{-2\ell}\Gamma(\ell), \qquad E_\ell^{(2)} \sim -\frac{\pi^5}{1152}\pi^{-2\ell}\Gamma(\ell + 1).$$

(3.15)

These predictions from the conjectural asymptotics can be tested against our calculation of the coefficients $E_\ell^{(k)}$. As an illustration, we show in Fig. 3 the first 30 terms of the sequence

$$S_\ell^{(k)} = \pi^{2\ell}\frac{E_\ell^{(k)}}{\Gamma(\ell + k - 1)},$$

(3.16)

as well as its second Richardson transform, for $k = 2$. They clearly converge rapidly to the expected value, $-\pi^5/1152$, represented by the horizontal line in the plot. We note that the study of the sequences $E_\ell^{(k)}$ suggests that the function $c(n)$ in (3.8) is given by

$$c(n) = -\frac{4}{\pi^3}\sin^2\left(\frac{\pi n}{2}\right).$$

(3.17)

Additional support for the conjectural large order behavior (3.8) can be obtained by studying the opposite limit, in which $n$ is close to 1. The expansion of the energy around $n = 1$ was studied in [33] (see also [34]), and one finds

$$E(u, n) = E(u, 1) + \frac{\pi}{2}\frac{I_1\left(\frac{\pi}{u}\right)}{I_0\left(\frac{\pi}{u}\right)}(1 - n)^2 + \mathcal{O}((1 - n)^4),$$

(3.18)

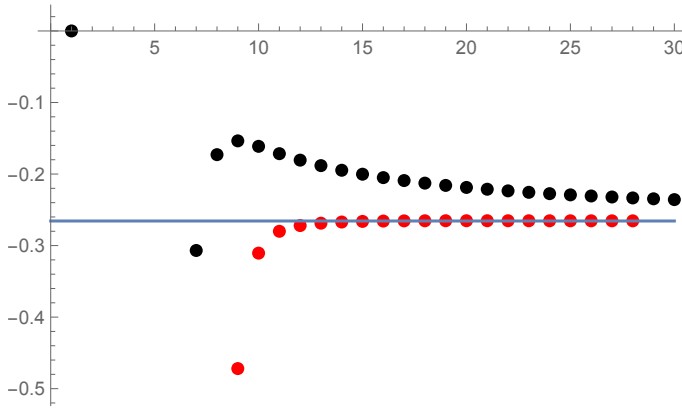

**Figure 3**. In this figure we show the first thirty points of the sequence $S_\ell^{(2)}$, defined in (3.16) (upper line of dots), together with its second Richardson transform (lower line of dots). The horizontal line is the expected asymptotic value $-\pi^5/1152$.

where $I_{0,1}(z)$ are Bessel functions. This leads to the following expansion for the coefficients of (2.14),

$$E_\ell(n) = E_\ell(1) + (1-n)^2 \widetilde{E}_\ell^{(1)} + \cdots , \qquad (3.19)$$

where the coefficients $\widetilde{E}_\ell^{(1)}$ are obtained by expanding the quotient of Bessel functions in (3.18). The prediction for the asymptotic behavior of this series at large $\ell$ can be obtained by expanding (3.8) around $n = 1$, and one finds

$$\widetilde{E}_\ell^{(1)} \sim -(2\pi)^{-\ell}\Gamma(\ell), \qquad \ell \gg 1. \qquad (3.20)$$

This can be again verified numerically.

We have then offered non-trivial evidence that the two-component, attractive Hubbard model at arbitrary filling satisfies the conjecture put forward in [5]: the perturbative series is factorially divergent and non-Borel summable, and the location of the first Borel singularity is determined by the squared energy gap.

### 3.3 Renormalons

Factorial growth in perturbation theory can be due to two different reasons. First of all, the total number of Feynman diagrams grows factorially with the loop order. This is expected to be captured by instantons [35–37]. In addition, there might be special sequences of diagrams, the so-called renormalon diagrams, which diverge factorially after integration over momenta [10], and lead to additional Borel singularities. When present, renormalon diagrams typically give the most important contribution to the large order behavior of the perturbative series (this has been verified in Yang–Mills theory [38] and in many two-dimensional asymptotically free theories [26, 28, 39, 40]). Diagramatically, renormalons are usually detected in some large $N$ analysis, where perturbation theory is dominated by a special family of diagrams. If this family of diagrams is of the renormalon type, one has established the existence of renormalons in the theory. Very often, the Borel singularity obtained in this way is the large $N$ approximation to the leading Borel singularity.

What is the source of the factorial growth that we have just found in the series (2.14) when $\kappa = 2$? As in the Gaudin–Yang model studied in [5], this growth is due to renormalon diagrams. A relevant renormalon sequence can be found by taking the limit in which the number of components $\kappa$ is very large. In this case, as we reviewed in section 2.1, ring diagrams give the leading non-trivial contribution to the ground state energy density. It turns out that these diagrams are of the renormalon type. More precisely, the contribution of ring diagrams to the ground state energy density at order $\ell$ grows with $\ell$ as

$$E_\ell^{\mathrm{ring}}(n; \kappa) \sim \left( \frac{4\pi}{\kappa} \sin\left( \frac{\pi \widetilde{n}}{2} \right) \right)^{-\ell} \ell!, \qquad \ell \gg 1. \tag{3.21}$$

This expression is valid when $\widetilde{n} < 1$. We note, first of all, that for $\kappa = 2$ this agrees with (3.8) at leading order (i.e. it leads to the $A(n)$ in (3.7)). Physically, the factorial growth in (3.21) is due to the logarithmic divergence of the polarization loop $\Pi(q, i\omega)$ at

$$\omega = 0, \qquad q = \pm 2k_F. \tag{3.22}$$

The asymptotic behavior (3.21) can be derived by looking carefully at the behavior of the integral near the singularity at (3.22). In section 4 we will establish (3.21) in a much more precise way, by calculating the trans-series associated to this sequence of diagrams. In the limit $n \to 0$, (3.21) agrees with the asymptotic behavior of ring diagrams in the multi-component Gaudin–Yang model [5].

We should note that, at half-filling $\widetilde{n} = 1$, ring diagrams have a different rate of divergence, given by

$$\left( \frac{2\pi}{\kappa} \right)^{-\ell} \ell! \tag{3.23}$$

We will also establish this in section 4. At least when $\kappa = 2$, this overestimates the growth of the perturbative series by a factor of 2: as already shown in [5], the exact perturbative result (2.30) leads to a growth

$$(2\pi)^{-\ell} \ell! \tag{3.24}$$

Although we have focused on ring diagrams, which dominate in the large $\kappa$ limit, at finite $\kappa$ there might be other families of diagrams with a similar renormalon behavior. For example, ladder diagrams are also expected to grow factorially with the loop order [5]. At half-filling and for $\kappa = 2$, the contribution of additional diagrams is even necessary to obtain the correct asymptotic growth (3.24). It would be interesting to understand this in more detail.

The diagrammatic analysis of renormalon singularities in a large $N$ limit makes it possible to identify some aspects of the non-perturbative effects, but it is clearly useful to have methods which do not rely on the details of the diagrammatics. In relativistic, asymptotically free theories, a renormalization group analysis (RG) determines the exponent $A$ in (3.3), as well as the sub-leading correction encoded in the exponent $b$, in terms of the beta function [41] (see also [25, 42]). This goes as follows. Let us assume that we have a running coupling constant $g(k)$, depending on a scale $k$, and satisfying a RG equation of the form

$$k \frac{\mathrm{d}g}{\mathrm{d}k} = \beta(g) = \beta_0 g^2 + \beta_1 g^3 + \cdots, \tag{3.25}$$

where $\beta_0 < 0$. Then, the following quantity

$$\mathcal{I} = k \, g(k)^{\beta_1/\beta_0^2} \mathrm{e}^{\frac{1}{\beta_0 g(k)}} \exp\left\{ -\int_{g_*}^{g(k)} \left( \frac{1}{\beta(\overline{g})} - \frac{1}{\beta_0 \overline{g}^2} + \frac{\beta_1}{\beta_0^2 \overline{g}} \right) \mathrm{d}\overline{g} \right\} \tag{3.26}$$

is invariant under the RG flow, i.e. it is independent of the scale $k$. Here $g_\star$ is an arbitrary value, which is equivalent to the freedom of multiplying $\mathcal{I}$ by an arbitrary $k$-independent constant. Since $\beta_0 < 0$, $\mathcal{I}$ is an exponentially small quantity in the coupling constant, and renormalon effects are believed to be roughly of the form $\mathcal{I}^d$, for an appropriate value of $d \in \mathbb{Z}_{\geq 0}$ (in relativistic QFT, $d$ is related to the dimension of the condensate responsible for the non-perturbative effect).

In the case of the two-component Hubbard model, the RG equation is obtained by integrating out degrees of freedom with wave vectors whose distance to the Fermi surface is larger than $k$ (see [43] for a review and references). An analysis of the non-perturbative effects associated to the RG flow in this case was made in [23]. The natural dimensionless coupling is

$$g = \frac{2u}{\pi v_F}, \qquad v_F = 2\sin\left(\frac{\pi n}{2}\right), \tag{3.27}$$

and the beta function has

$$\beta_0 = -1, \qquad \beta_1 = \frac{1}{2}. \tag{3.28}$$

The energy gap scales as $\mathcal{I}$, and one finds in this way

$$\Delta \approx u^{1/2} \exp\left(-\frac{\pi}{u}\sin\left(\frac{\pi n}{2}\right)\right), \tag{3.29}$$

which agrees with the Bethe ansatz result (2.50) (it is actually possible to recover the full Bethe ansatz answer, including numerical prefactors). In the case of the ground state energy, the corresponding non-perturbative effect scales like the square of the gap, i.e. $d = 2$.

We conclude that, as in relativistic QFT, the RG analysis gives information about renormalons. In particular, the location of the Borel singularity is determined by the first coefficient of the beta function and the "dimension" $d$. At the same time, this information can be retrieved from the large order behavior of the perturbative series.

## 3.4 A resurgent prediction for the gap in the multi-component case

The conjecture of [5] makes it in principle possible to obtain the form of the energy gap from an analysis of the large order behavior of the perturbative expansion. In the Gaudin–Yang model and the two-component Hubbard model, the energy gap is known thanks to the Bethe ansatz. However, the multi-component, attractive Hubbard model is not integrable. From a Luttinger liquid analysis [44, 45], it is known that the spin sector has a gap, due to the formation of bound states of $\kappa$ fermions. As far as we know, the asymptotic form of the gap at weak coupling has not been determined. In this section we will put various results together in order to obtain a proposal for the energy gap in that case, i.e. for the functions $A(n; \kappa)$ and $b(n; \kappa)$ in (3.6).

First of all, the exponent of the gap for the multicomponent case can be obtained by looking at the factorial divergence of ring diagrams, since renormalon diagrams are expected to give the correct location of the dominating Borel singularity. Therefore, we have

$$A(n; \kappa) = \frac{4\pi}{\kappa}\sin\left(\frac{\pi n}{\kappa}\right). \tag{3.30}$$

This expression gives the correct answer (3.7) for $\kappa = 2$ and arbitrary $n$. In the limit $n \to 0$ it also reproduces the correct answer for the Gaudin–Yang model with $\kappa$ components.

Next, we consider the exponent $b(n; \kappa)$. We know that it is independent of $n$ in the case $\kappa = 2$. Assuming that this continues to be the case for general $\kappa$, we can determine it by

considering the limit of the multi-component Gaudin–Yang model. A large order analysis of the perturbative series for the energy $e_{\mathrm{GY}}(\lambda; \kappa)$ in (2.17) shows that

$$e_\ell(\kappa) \sim \left(-\pi^2\right)^{-\ell} \Gamma\left(\ell - \frac{2}{\kappa}\right). \tag{3.31}$$

This gives $b(n; \kappa) = -2/\kappa$. We have to take now into account that the non-perturbative scale associated to the energy is the square of the one appearing in the energy gap. We then conclude that this gap, in the multi-component Hubbard model, is of the form

$$\Delta \approx u^{1/\kappa} \exp\left(-\frac{2\pi}{\kappa u} \sin\left(\frac{\pi n}{\kappa}\right)\right), \qquad u \to 0. \tag{3.32}$$

This agrees with the result (2.50) when $\kappa = 2$. Physically, the gap is due to the formation of bound states of $\kappa$ particles, generalizing the familiar superconducting gap in the case $\kappa = 2$. It characterizes the "molecular superfluid" phase of the multi-component Hubbard model in one dimension [44, 45][3]. It would be very interesting to test the conjecture (3.32) with other techniques. From the point of view of the RG, this gap would be generated by a beta function with the following coefficients:

$$\beta_0 = -\frac{\kappa}{2}, \qquad \beta_1 = \frac{\kappa}{4}. \tag{3.33}$$

We note that (3.32) compares well with the coupling constant dependence of the non-perturbative scale in the $SU(\kappa)$ chiral Gross–Neveu model, which can be regarded as the continuum limit of the multi-component Hubbard model [47–49][4]. In that case, one can use the beta function at two-loops (see e.g. [50]) to find that the RG-invariant scale is given by

$$\Lambda \approx g^{1/\kappa} \exp\left(-\frac{1}{\kappa g}\right). \tag{3.34}$$

This has the form (3.32), where the coupling $g$ is given by (3.27), and one uses $\widetilde{n}$ instead of $n$ in the Fermi velocity.

## 4 Resurgent analysis of the ground state energy density

In this section we will perform a detailed analysis of some of the perturbative series that we have studied, by using the tools of the theory of resurgence. For an elementary survey of these tools we refer the reader to [1, 25, 51]. A more comprehensive review can be found in [3].

### 4.1 Tools from resurgent analysis

Let us first review some of the basic ingredients we will need for a deeper analysis of the perturbative series. These series are factorially divergent and their Borel transforms (3.4) have singularities on the positive real axis. This means that the conventional Borel resummation

$$s(\varphi)(z) = z^{-1} \int_0^\infty \mathrm{d}\zeta\, \mathrm{e}^{-\zeta/z} \widehat{\varphi}(\zeta) \tag{4.1}$$

---

[3]A detailed derivation of the gap in the multi-component Gaudin–Yang model, directly from the Bethe ansatz equations, is presented in [46]. It is in perfect agreement with the large order behavior (3.31) and the conjecture of [5].

[4]We would like to thank Philippe Lecheminant for suggesting this comparison.

is ill-defined. However, one can define the so-called *lateral Borel resummations* as

$$s_\pm(\varphi)(z) = z^{-1} \int_{\mathcal{C}_\pm} \mathrm{d}\zeta \, \mathrm{e}^{-\zeta/z} \widehat{\varphi}(\zeta), \tag{4.2}$$

where $\mathcal{C}_\pm$ are integration paths slightly above (respectively, below) the positive real axis. If all the coefficients of the original perturbative series are real, the lateral Borel resummations have an imaginary piece. This is captured by the *Stokes automorphism* or *Stokes discontinuity*,

$$\mathrm{disc}(\varphi)(z) = s_+(\varphi)(z) - s_-(\varphi)(z). \tag{4.3}$$

The Stokes discontinuity is sensitive to the singularities of the Borel transform $\widehat{\varphi}(\zeta)$, which obstruct conventional Borel summability. Let us suppose that $\widehat{\varphi}(\zeta)$ has a singularity at $\zeta = A$. Then, the discontinuity has an asymptotic expansion, for $z$ small, of the form

$$\mathrm{disc}(\varphi)(z) \sim \mathrm{i} \, \mathrm{e}^{-A/z} z^{-b} \sum_{k=0}^{\infty} \mu_k z^k. \tag{4.4}$$

The values of $b$ and the $\mu_n$ depend on the behavior of $\widehat{\varphi}(z)$ near the singularity. The r.h.s. of (4.4) involves two small parameters: $z$ and $\mathrm{e}^{-A/z}$. For this reason, it is an example of a *trans-series*, which incorporate explicitly exponentially small effects. If there are singularities at positive integer multiples of $A$ (as it is often the case), the trans-series associated to them has the general form

$$\sum_{\ell=1}^{\infty} C_\ell \, \mathrm{e}^{-\ell A/z} z^{-b_\ell} \varphi_\ell(z), \tag{4.5}$$

where $C_\ell$ are constants (sometimes called trans-series parameters), and

$$\varphi_\ell(z) = \sum_{k \geq 0} a_{\ell,k} z^k \tag{4.6}$$

are formal power series in $z$. In general they are factorially divergent series, i.e. $a_{\ell,k} \sim k!$ for fixed $\ell$[5]

As an example of trans-series which will be relevant for the Hubbard model, let us suppose that the singularity of $\widehat{\varphi}(\zeta)$ near $\zeta = A$ involves a pole and a logarithmic branch cut. We then have

$$\widehat{\varphi}(A + \xi) = -\frac{a}{\xi} - \log(\xi) \sum_{k \geq 0} \hat{c}_k \xi^k + \cdots \tag{4.7}$$

A simple calculation (see e.g. [25], section 3.2) shows that the Stokes discontinuity due to this singularity will have the asymptotic expansion, for $z$ small,

$$\mathrm{disc}(\varphi)(z) \sim 2\pi \mathrm{i} \, \mathrm{e}^{-A/z} \left( \frac{a}{z} + \sum_{k \geq 0} c_k z^k \right), \tag{4.8}$$

---

[5]Note that the parameters $C_\ell$ are only well defined once a normalization has been chosen for $\varphi_\ell(z)$, which is what we will do here. When a trans-series of the form (4.5) is obtained from the solution of an ordinary differential equation, one has a further constraint that $C_\ell = C^\ell$, for some $C$, but in QFT problems one might need the more general ansatz (4.5).

where

$$c_k = \Gamma(k+1)\hat{c}_k. \tag{4.9}$$

Physically, trans-series encode information about non-perturbative effects of the theory. Conversely, exponentially small corrections like (3.3) should be regarded as approximations to trans-series. In many examples, these effects are of the instanton type (as in for example one-dimensional quantum mechanics). In other examples, these effects are induced by renormalon singularities and they are associated to other types of non-perturbative physics (in asymptotically free, relativistic quantum field theories, it has been proposed that they encode information about non-perturbative condensates, see e.g. [10] and references therein).

Another important aspect of trans-series is that they determine the large order behavior of the coefficients of the original, perturbative series. It is easy to show, by using a dispersion relation, that (4.4) implies

$$a_k \sim \frac{1}{2\pi} A^{-b-k} \sum_{r=0}^{\infty} \mu_r A^r \Gamma(k+b-r), \qquad k \gg 1. \tag{4.10}$$

This is the the all-orders generalization of (3.2).

In this section we will extract the trans-series associated to the ground state energy density of the Hubbard model. First, we will do an approximate calculation by extracting the trans-series from ring diagrams. This is the dominant contribution at large $\kappa$, and it is the analogue of the large $\beta_0$ approximation familiar in studies of renormalons in QED and QCD [10]. However, in the half-filled case, the explicit expression of Misurkin–Ovchinnikov [11] for the perturbative series will make it possible to determine the *exact* trans-series, as we will see.

## 4.2   Trans-series and ring diagrams

As we have seen in section 3.3, the ring diagrams of the one-dimensional Hubbard model are renormalon diagrams, which grow factorially with the loop order and give information on the location of the dominant Borel singularity. We will now show that the trans-series associated to this sequence of diagrams can be computed systematically. To do this, we will use a trick introduced in [52].

As it is well-known, the formal series of ring diagrams

$$E^{\mathrm{ring}}(n;\kappa) = \sum_{\ell \geq 2} E_\ell^{\mathrm{ring}}(n;\kappa) u^\ell \tag{4.11}$$

can be resummed in terms of the integral of a logarithm

$$E^{\mathrm{ring}}(n;\kappa) = \int_{-\pi}^{\pi} \frac{\mathrm{d}q}{2\pi} \int_{\mathbb{R}} \frac{\mathrm{d}\omega}{2\pi} \left\{ -v\Pi(q,\mathrm{i}\omega) + \frac{1}{2}\log\left(1 + 2v\Pi(q,\mathrm{i}\omega)\right) \right\}. \tag{4.12}$$

This integral develops an imaginary part when the argument of the logarithm becomes negative. The asymptotic expansion of this imaginary part, for $v$ small, gives the trans-series associated to the perturbative series of ring diagrams (4.11). It turns out that the we have to distinguish explicitly the case $\widetilde{n} < 1$ from the half-filling case $\widetilde{n} = 1$.

Let us start with the case $\widetilde{n} < 1$. We first note that, due to the symmetry,

$$\Pi(q,\mathrm{i}\omega) = \Pi(-q,\mathrm{i}\omega) = \Pi(q,-\mathrm{i}\omega) = \Pi(-q,-\mathrm{i}\omega), \tag{4.13}$$

we can focus on the upper right quadrant of the $q, \omega$ plane. The imaginary part of (4.12) is given by the integral

$$\Sigma(n; \kappa) = \frac{1}{\pi} \int_0^\pi \mathrm{d}q \int_0^\infty \mathrm{d}\omega \, \Theta \left( -(1 + 2\upsilon \Pi(q, \mathrm{i}\omega)) \right) = \frac{1}{\pi} \int_{q_-}^{q_+} \mathrm{d}q \, Q_0(q). \tag{4.14}$$

Here we have introduced

$$- \Pi(q_\pm, 0) = \frac{1}{2\upsilon}, \quad -\Pi(q, \mathrm{i}Q_0(q)) = \frac{1}{2\upsilon} \tag{4.15}$$

and $\Theta(x)$ is the Heaviside function. To proceed, we introduce the trans-series parameter

$$\alpha = \exp\left( -\frac{2\pi}{\upsilon} \sin\left( \frac{\pi \widetilde{n}}{2} \right) \right), \tag{4.16}$$

as well as the following variables

$$w = \frac{1}{2\alpha} \left( \frac{\sin(q/2)}{\sin(\pi \widetilde{n}/2)} - 1 \right), \quad \nu = \alpha^{-2} \left( \sqrt{\frac{1}{16} Q_0^2 \csc^2\left( \frac{q}{2} \right) + 1} - 1 \right). \tag{4.17}$$

Let us define $w_\pm = w(q_\pm)$ as the solutions to

$$w_\pm \mp \mathrm{e}^{2\alpha \log \alpha w_\pm} (1 + \alpha w_\pm) = 0. \tag{4.18}$$

They can be obtained as a power series in $\alpha$:

$$w_\pm = \pm 1 + \alpha(2 \log(\alpha) + 1) \pm \alpha^2 \left( 6 \log^2(\alpha) + 6 \log(\alpha) + 1 \right) + \mathcal{O}(\alpha^3). \tag{4.19}$$

We now write

$$\frac{\nu}{4} \left( \left( \alpha^2 \nu + 2 \right) \csc^2 \left( \frac{\pi \widetilde{n}}{2} \right) \left( 1 - \alpha^2 \mathrm{e}^{2\alpha \log \alpha \left( 2w + \alpha \nu + 2\alpha^2 \nu w \right)} \right) \right.$$
$$\left. -2(2\alpha w + 1) \left( 1 + \alpha^2 \mathrm{e}^{2\alpha \log \alpha \left( 2w + \alpha \nu + 2\alpha^2 \nu w \right)} \right) \right) = g_+(w, \nu) g_-(w, \nu), \tag{4.20}$$

where

$$g_\pm(w, \nu) = (w - w_\pm) \pm \mathrm{e}^{2\alpha \log \alpha w_\pm} \left( 1 + \alpha w_\pm \left( 1 - \mathrm{e}^{2\alpha \log \alpha (w - w_\pm)} \right) \right.$$
$$\left. - \mathrm{e}^{2\alpha \log \alpha (w - w_\pm)} \left( \alpha w_\pm - (1 + \alpha w) \mathrm{e}^{\alpha^2 \log \alpha \nu (2\alpha w + 1)} \right) \right). \tag{4.21}$$

We solve (4.20) to find $\nu(w)$ as function of $w$ in a power series in $\alpha$. For the first few orders, we obtain

$$\nu(w) = 2 \tan^2 \left( \frac{\pi \widetilde{n}}{2} \right) (w - w_-)(w_+ - w) + 4\alpha \left( w (w - w_-)(w_+ - w) \tan^4 \left( \frac{\pi \widetilde{n}}{2} \right) \right)$$
$$+ \mathcal{O}\left( \alpha^2 \right). \tag{4.22}$$

We are finally in place to perform the integral

$$\frac{1}{\pi} \int_{q_-}^{q_+} \mathrm{d}q \, Q_0(q)$$

$$= \frac{1}{\pi} \int_{w_-}^{w_+} \frac{4\alpha \sin\left( \frac{\pi \widetilde{n}}{2} \right)}{\sqrt{1 - \sin^2\left( \frac{\pi \widetilde{n}}{2} \right) (2\alpha w + 1)^2}} \left( 4 \sin\left( \frac{\pi \widetilde{n}}{2} \right) (1 + 2\alpha w) \sqrt{(1 + \alpha^2 \nu)^2 - 1} \right) \mathrm{d}w, \tag{4.23}$$

by using again an expansion in $\alpha$. One finds the trans-series,

$$\Sigma(n;\kappa) = 16\,\mathrm{e}^{-\frac{4\pi}{v}\sin\left(\frac{\pi\widetilde{n}}{2}\right)}\sin\left(\frac{\pi\widetilde{n}}{2}\right)\tan^2\left(\frac{\pi\widetilde{n}}{2}\right)$$

$$+\,\mathrm{e}^{-\frac{8\pi}{v}\sin\left(\frac{\pi\widetilde{n}}{2}\right)}\left(\frac{2048\pi^2\sin^7\left(\frac{\pi\widetilde{n}}{2}\right)\csc^2(\pi\widetilde{n})}{v^2} - \frac{64\left(\pi(\cos(\pi\widetilde{n})+5)\tan^4\left(\frac{\pi\widetilde{n}}{2}\right)\right)}{v}\right) \quad (4.24)$$

$$+16(\cos(\pi\widetilde{n})+3)\tan^3\left(\frac{\pi\widetilde{n}}{2}\right)\sec^3\left(\frac{\pi\widetilde{n}}{2}\right)\Bigg) + \mathcal{O}\left(\mathrm{e}^{-\frac{12\pi}{v}\sin\left(\frac{\pi\widetilde{n}}{2}\right)}\right).$$

This has the general form in (4.5) with

$$A = \frac{4\pi}{\kappa}\sin\left(\frac{\pi\widetilde{n}}{2}\right), \qquad b_\ell = 2(\ell-1). \quad (4.25)$$

In particular, this trans-series indicates the existence of singularities in the Borel plane of $u$ at the locations

$$\zeta = \frac{4\pi\ell}{\kappa}\sin\left(\frac{\pi\widetilde{n}}{2}\right), \qquad \ell \in \mathbb{Z}_{>0}. \quad (4.26)$$

In addition, the series $\varphi_\ell(u)$ truncates to a polynomial of degree $2(\ell-1)$. This seems to be typical of approximations based on families of "bubble"-like diagrams [52]. By using the connection between non-perturbative effects and large order behavior, we verify in particular the factorial growth (3.21).

The above result is not valid in the case of half-filling, $\widetilde{n} = 1$, and we have to do a different analysis. We introduce the parameter

$$\alpha = \mathrm{e}^{-\frac{2\pi}{v}} \quad (4.27)$$

and we make the following change of variables:

$$s = \frac{1}{\alpha}\left(1 - \sin\left(\frac{q}{2}\right)\right), \quad t = \alpha^{-1}\left(\sqrt{\frac{1}{16}Q_0^2\csc^2\left(\frac{q}{2}\right)+1} - 1\right). \quad (4.28)$$

In this case, the upper limit of the integral in (4.14) is always $q = \pi$. For the limit $s_- = s(q_-)$, we have

$$s_- = \mathrm{e}^{-\frac{2\pi}{v}s_-}(2 - \alpha s_-). \quad (4.29)$$

We can now solve for $t$

$$t = \frac{(s_- - s) + \mathrm{e}^{\frac{2\pi\alpha s_-}{v}}(\alpha s_- - 2) + \mathrm{e}^{\frac{2\pi\alpha(\alpha s t + s - t)}{v}}(2 - \alpha s)}{1 - \alpha\,\mathrm{e}^{\frac{2\pi\alpha(\alpha s t + s - t)}{v}}}. \quad (4.30)$$

which can be easily expanded in powers of $\alpha$ with the result

$$t(s) = (s_- - s)\left\{1 + \alpha\left(2 - \frac{8\pi}{v}\right) + \alpha^2\left(2 + \frac{4\pi(3s-4)}{v} - \frac{16\left(\pi^2(s-2)\right)}{v^2}\right) + \mathcal{O}\left(\alpha^3\right)\right\}. \quad (4.31)$$

The trans-series is then given by the integral

$$\Sigma(u) = \frac{1}{\pi}\int_0^{s_-} 8\alpha(1 - \alpha s)\sqrt{\frac{t\,(2 + \alpha t)}{s\,(2 - \alpha s)}}\,\mathrm{d}s, \quad (4.32)$$

which can be expanded as follows:

$$\Sigma(u) = 8\,\mathrm{e}^{-\frac{2\pi}{v}} + \mathrm{e}^{-\frac{6\pi}{v}}\left(\frac{96\pi^2}{v^2} - \frac{64\pi}{v} + 8\right)$$
$$+ \mathrm{e}^{-\frac{10\pi}{v}}\left(\frac{4000\pi^4}{v^4} - \frac{4800\pi^3}{v^3} + \frac{1760\pi^2}{v^2} - \frac{224\pi}{v} + 8\right) + \mathcal{O}\left(\mathrm{e}^{-\frac{14\pi}{v}}\right). \tag{4.33}$$

The structure of singularities is now different. They are located at odd integer multiples of $2\pi/\kappa$ with

$$\zeta = \frac{2\pi(2r+1)}{\kappa}, \qquad r \in \mathbb{Z}_{\geq 0}. \tag{4.34}$$

As we mentioned in section 3.3, when $\kappa = 2$ this does not agree with the exact result, which can be obtained from (2.30) (see (4.54)). This is in contrast with the case $\widetilde{n} < 1$, and seems to be an idiosyncrasy of the model at half-filling. It should however be a warning on the use of specific classes of factorially divergent diagrams in order to locate Borel singularities.

### 4.3 Trans-series at half-filling and the Stokes automorphism

In the half-filled case, the perturbative series for the energy is explicitly given by (2.30). We will denote by

$$\Phi(u) = \sum_{k=0}^{\infty} h_{k+1} u^{2k} \tag{4.35}$$

the associated formal power series. The exact ground state energy density (2.48) has the asymptotic expansion

$$E(u,1) \sim -\frac{4}{\pi} - \frac{u}{2} - u^2 \Phi(u). \tag{4.36}$$

We want to study the resurgent properties of this series. The first step is to analyze the Borel transform of the formal power series $\Phi(u)$. We can separate the coefficients $h_k$ into two parts:

$$h_k = h_k^{(1)} h_k^{(2)}, \qquad k \geq 1, \tag{4.37}$$

where

$$h_k^{(1)} = \left(\frac{\Gamma(2k-1)}{\pi^3(2\pi)^{2k-1}}\right)\left(\frac{4(2k-1)\Gamma(k-1/2)^2}{\Gamma(k)^2}\right) \tag{4.38}$$

and

$$h_k^{(2)} = (1 - 2^{-2k-1})\zeta(2k+1). \tag{4.39}$$

The second factor can be written as

$$(1 - 2^{-2k-1})\zeta(2k+1) = (1 - 2^{-2k-1})\left(1 + \sum_{j=2}^{\infty} \mathrm{e}^{-(2k+1)\log j}\right) = \sum_{j=0}^{\infty} \mathrm{e}^{-(2k+1)\log(2j+1)}, \tag{4.40}$$

and as we will see, it leads to additional singularities along the positive real axis. Let us then focus on the formal power series associated to $h_k^{(1)}$:

$$\varphi(u) = \sum_{k=0}^{\infty} h_{k+1}^{(1)} u^{2k}. \tag{4.41}$$

Its Borel transform can be computed in closed form as

$$\widehat{\varphi}(\zeta) = \sum_{k=0}^{\infty} \frac{h_{k+1}^{(1)}}{(2k)!} \zeta^{2k} = \frac{16}{\pi^2} \frac{1}{4\pi^2 - \zeta^2} E\left(\frac{\zeta^2}{4\pi^2}\right), \tag{4.42}$$

where $E(k^2)$ is the complete elliptic integral of the second kind, which can be also written as a hypergeometric function:

$$E(z) = \frac{\pi}{2} \, {}_2F_1\left(-\frac{1}{2}, \frac{1}{2}, 1; z\right). \tag{4.43}$$

The Borel transform (4.42) has a singularity at $\zeta = 2\pi$, where there is a pole and a branch cut. Near the singularity, we have

$$\widehat{\varphi}(2\pi + \xi) = -\frac{a}{\xi} - \log(\xi) \sum_{n \geq 0} \hat{c}_n \xi^n + \cdots \tag{4.44}$$

where

$$a = \frac{4}{\pi^3} \tag{4.45}$$

and the coefficients $\hat{c}_n$ can be computed in closed form by using the discontinuity of the hypergeometric function:

$${}_2F_1\left(-\frac{1}{2}, \frac{1}{2}, 1; z + i\epsilon\right) - {}_2F_1\left(-\frac{1}{2}, \frac{1}{2}, 1; z - i\epsilon\right) = (1 - z) \, {}_2F_1\left(\frac{3}{2}, \frac{1}{2}, 2; 1 - z\right), \qquad z > 1. \tag{4.46}$$

One finds,

$$\sum_{n \geq 0} \hat{c}_n \xi^n = \frac{1}{\pi^4} \, {}_2F_1\left(\frac{3}{2}, \frac{1}{2}, 2; -\frac{\xi(\xi + 4\pi)}{4\pi^2}\right). \tag{4.47}$$

It follows from (4.8) that

$$\mathrm{disc}(\varphi)(z) \sim i \, e^{-2\pi/z} z^{-1} \varphi_1(z), \tag{4.48}$$

where

$$\varphi_1(z) = \sum_{n \geq 0} \varphi_{1,n} z^n = 2\pi \left(a + \sum_{n \geq 0} c_n z^{n+1}\right) = \frac{8}{\pi^2} + \frac{2}{\pi^3} z - \frac{3}{4\pi^4} z^2 + \frac{9}{16\pi^5} z^3 + \cdots \tag{4.49}$$

The series $\varphi_1(z)$ is Borel summable along the positive real axis, and its Borel resummation can be written as:

$$s(\varphi_1)(z) = 2\pi a + \frac{2}{\pi^3} \int_0^{\infty} e^{-\zeta/z} \, {}_2F_1\left(\frac{3}{2}, \frac{1}{2}, 2; -\frac{\zeta(\zeta + 4\pi)}{4\pi^2}\right) d\zeta, \tag{4.50}$$

where $z > 0$.

Let us now come back to the original series $\Phi(u)$. By using the representation (4.40), one finds

$$\widehat{\Phi}(\zeta) = \sum_{j=0}^{\infty} \frac{1}{2j+1} \widehat{\varphi}\left(\frac{\zeta}{(2j+1)}\right). \tag{4.51}$$

This is purely formal since the series in the r.h.s. is not convergent. However, it leads to the correct equation for the Stokes discontinuity of $\Phi(z)$:

$$\text{disc}(\Phi)(z) \sim \frac{1}{z} \sum_{r=0}^{\infty} e^{-2\pi(2r+1)/z} \varphi_{2r+1}(z). \tag{4.52}$$

where

$$\varphi_\ell(z) = \frac{1}{\ell^2} \varphi_1 \left( \frac{z}{\ell} \right). \tag{4.53}$$

(4.52) has an important physical interpretation. It says that the Borel transform of $\Phi$ has singularities on the positive real axis at odd, integer multiples of $2\pi$:

$$\zeta = (2r+1)2\pi, \qquad r = 0, 1, \cdots, \tag{4.54}$$

which we represent in Fig. 4 (there are also singularities at the reflected points $-(2r+1)2\pi$ on the negative real axis). Each of these singularities corresponds to a non-perturbative sector. The formal power series $\varphi_{2r+1}(z)$ represent the quantum fluctuations in this sector. If these were instanton sectors, $\varphi_{2r+1}(z)$ would give the expansion of the path integral around a non-trivial saddle point. In this case we do not have a semiclassical interpretation of these sectors, but we can compute their associated trans-series explicitly.

It is interesting to note that the structure of Borel singularities in the ground state energy density, located at odd integer multiples of $2\pi$, is due to the presence of the zeta function $\zeta(2k+1)$ in the perturbative term of order $k$, as it is clear from (4.40). It is tempting to think that the zeta functions appearing recurrently in perturbative expansions in quantum theory are giving information on the underlying trans-series structure. One should also note that the structure of (4.52) is very similar to the one appearing in the exact gap (2.51) for the half-filled case, which is given as well by an infinite series of exponentials. The corresponding singularities are at odd integer multiples of $\pi$, instead of $2\pi$. This is of course in line with the fact that the ground state energy density has the Borel structure of the square of the gap.

It is possible to derive the formal trans-series appearing in the r.h.s. of (4.52) in a different way, by using the integral representation of the exact energy in [11]. We sketch this derivation in Appendix B.

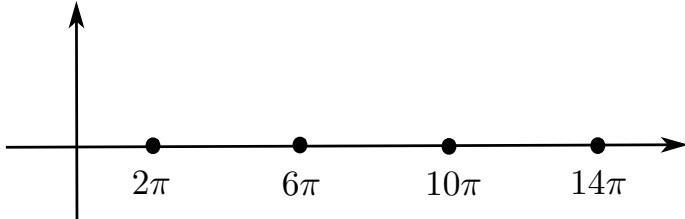

**Figure 4**. The very first singularities of the Borel transform of $\Phi(z)$ on the positive real axis.

The asymptotic equality (4.52) can be promoted to an *exact* discontinuity formula by Borel resumming the r.h.s. We obtain in this way, when $z > 0$,

$$s_+(\Phi)(z) - s_-(\Phi)(z) = \frac{1}{z} \sum_{r=0}^{\infty} e^{-2\pi(2r+1)/z} s(\varphi_{2r+1})(z). \tag{4.55}$$

We note that the formal power series $\varphi_\ell(z)$ are all Borel summable along the positive real axis (this follows from (4.53) and the Borel summability of $\varphi_1(z)$ for $z > 0$). Exact discontinuity formulae of this type are rare in quantum theory. One exception is the discontinuity of Voros symbols in the exact WKB method, which are given by the so-called Delabaere–Pham formula [53]. In the Delabaere–Pham formula, the discontinuity is also given by an infinite sum of exponentially small functions, which can be resummed into a logarithm involving only $\varphi_1(z)$. The structure of (4.55) is different, and in particular it does not admit a resummation (although all the series $\varphi_{2r+1}(z)$ can be obtained from $\varphi_1(z)$, as shown in (4.53)).

The exact discontinuity formula (4.55) has some interesting consequences. Let us define the discontinuity function as

$$\mathcal{D}(u) = \sum_{r=0}^{\infty} e^{-2\pi(2r+1)/u} s(\varphi_{2r+1})(u). \tag{4.56}$$

We can use the Borel transform (4.50) to represent it as a single integral

$$\begin{aligned}
\mathcal{D}(u) = {} & \frac{8}{\pi^2} \left( \text{Li}_2\left(e^{-2\pi/u}\right) - \frac{1}{4}\text{Li}_2\left(e^{-4\pi/u}\right) \right) \\
& + \frac{2}{\pi^3} \int_0^\infty \left( \text{Li}_2\left(e^{-(2\pi+\zeta)/u}\right) - \frac{1}{4}\text{Li}_2\left(e^{-2(2\pi+\zeta)/u}\right) \right) \, {}_2F_1\left(\frac{3}{2}, \frac{1}{2}, 2; -\frac{\zeta(\zeta+4\pi)}{4\pi^2}\right) \mathrm{d}\zeta.
\end{aligned} \tag{4.57}$$

We can think about (4.55) as a dispersion relation, relating the perturbative sector of the theory (represented by $\Phi(u)$) to the non-perturbative sector (represented by $\mathcal{D}(u)$). This is the basis for the resurgent formulae connecting the large order behavior of the coefficients of the perturbative series, to non-perturbative effects (see e.g. [25]). In this case, the exact discontinuity formula (4.55) leads to the following formula for the perturbative coefficients:

$$h_k = \frac{1}{\pi} \int_0^\infty \mathcal{D}(u) u^{-2k} \mathrm{d}u. \tag{4.58}$$

By performing the integral, one finds:

$$h_k = \frac{8}{\pi^3}(2\pi)^{-2k+1}\Gamma(2k-1)\left(1 + \mathcal{A}_k\right)(1 - 2^{-2k-1})\zeta(2k+1), \tag{4.59}$$

where

$$\mathcal{A}_k = \frac{1}{4\pi} \int_0^\infty \left(1 + \frac{\zeta}{2\pi}\right)^{1-2k} {}_2F_1\left(\frac{3}{2}, \frac{1}{2}, 2; -\frac{\zeta(\zeta+4\pi)}{4\pi^2}\right) \mathrm{d}\zeta. \tag{4.60}$$

Standard properties of hypergeometric functions lead to

$$\mathcal{A}_k = \frac{(2k-1)\Gamma\left(k - \frac{1}{2}\right)^2}{2\Gamma(k)^2} - 1. \tag{4.61}$$

Of course, by using this result, one can verify that (4.59) gives back the original expression (2.30). However, the expression (4.59) provides a "decoding" of these coefficients in terms of non-perturbative information, and it can be regarded as an *exact* large order formula. The first factor gives the leading larger order behavior

$$h_k \sim \frac{8}{\pi^3}(2\pi)^{-2k+1}\Gamma(2k-1), \qquad k \gg 1. \tag{4.62}$$

This contains information associated to the leading behavior of the trans-series $\varphi_1(z)$, namely the exponent and the prefactor, as in (3.2), (3.3). The factor $\mathcal{A}_k$ in (4.59) gives the corrections in $1/k$ to this leading behavior, involving the subleading terms in $\varphi_1(z)$, as in (4.10). Finally, the factor involving the zeta function, when written as in (4.40), provides exponentially small corrections in $k$ to the all-orders asymptotics in $1/k$, and contains the information about the remaining trans-series $\varphi_{2r+1}(z)$ with $r \geq 1$ (including the location of the subleading singularities).

It is natural to conjecture that the exact discontinuity formula (4.55) can be refined to the following expression for the lateral Borel resummations:

$$s_{\pm}(\Phi)(u) = \frac{1}{\pi} \int_0^{\infty} \frac{\mathcal{D}(z)}{z^2 - u^2 \pm \mathrm{i}\epsilon} \mathrm{d}z. \tag{4.63}$$

We have verified this conjecture numerically.

Finally, after this detailed study of the resurgent structure of the perturbative series of Misurkin–Ovchinnikov, we should address the question of how one recovers from it the exact ground state energy density, given in (2.48). The asymptotic expansion (4.36) leads to two different lateral Borel resummations, which we will denote as

$$\mathcal{E}_{\pm}(u) = -\frac{4}{\pi} - \frac{u}{2} - u^2 s_{\pm}(\Phi)(u). \tag{4.64}$$

These quantities are not real, and their imaginary parts are given by

$$\mathrm{Im}\, \mathcal{E}_{\pm}(u) = \pm \frac{u}{2} \mathcal{D}(u). \tag{4.65}$$

The physical energy $E(u, 1)$ is of course real. It turns out that $E(u, 1)$ can be obtained by the simplest possible resummation procedure leading to a real answer: the so-called *median resummation* [53]. In this case, since the trans-series $\varphi_{\ell}(z)$ in (4.53) are Borel summable along the positive real axis, the median resummation is even simpler and it is given by just half the sum of the two lateral Borel resummations:

$$E(u, 1) = \frac{1}{2} \left( \mathcal{E}_+(u) + \mathcal{E}_-(u) \right). \tag{4.66}$$

Although we have not proved this equality, we have verified it numerically with high precision. In this sense, the exact ground state energy density of the model does not require explicit non-perturbative information: one can reconstruct it by just using lateral Borel resummations of the perturbative expansions.

## 5 Conclusions and outlook

In this paper we have continued the program of [5] and we have studied the attractive, one-dimensional Hubbard model from the point of view of the theory of resurgence. In the two-component case, we have given evidence that the perturbative series for the ground state energy density is factorially divergent and not Borel summable, and that its Borel singularity is determined by the energy gap, for arbitrary filling. We have also shown that it is a renormalon singularity and we have identified a particular sequence of renormalon diagrams: the ring diagrams, similarly to what happens in the Gaudin–Yang model studied in [5]. By using all this information, we have proposed an explicit expression for the energy gap at weak coupling in the multi-component Hubbard model (at next-to-leading order in the coupling constant). We have

also clarified the connection between renormalons, energy gap, large order behavior, and the renormalization group.

In the case of half-filling, the very explicit results of Misurkin and Ovchinnikov [11] on the all-orders perturbative series make it possible to determine explicitly the exact trans-series and Stokes discontinuity for the ground state energy density. Similar exact results have been obtained in quantum mechanics, in the context of the exact WKB method, but they are scarce in quantum field theory (see [54] for a recent result in the same direction).

There are clearly many directions to explore and improve our results. Despite our efforts, we were not able to obtain the coefficients of the perturbative series (2.14) in closed form, as functions of the filling $n$, and we had to perform an expansion around $n = 0$. It would be very interesting to extend the techniques of [26, 27] to solve this problem. Our proposal for the energy gap in the multi-component case could be tested with RG techniques or with numerical simulations (a detailed study in the multi-component Gaudin–Yang model will appear in [46]). A natural generalization would be to consider one-dimensional Hubbard chains with other global symmetry groups (see e.g. [55, 56]). It is also clear that resurgent techniques could be applied fruitfully to similar systems, like the Kondo problem.

A more fundamental open problem would be to derive the trans-series from first principles. The lack of a semiclassical description of renormalons makes this a difficult problem. However, our explicit result for the exact trans-series at half-filling could be regarded as a signpost and a testing ground for new ideas on renormalons (see e.g. [57–63]).

Last but not least, it would be interesting to explore these ideas in the Hubbard model in higher dimensions. It is well known that, in the limit of infinite dimensionality, the model simplifies significantly, even at the diagrammatic level [64, 65]. A resurgent analysis in this regime might provide interesting insights on the model and on resurgent quantum theory in general.

## Acknowledgements

We would like to thank Thierry Giamarchi, Wilhelm Zwerger and specially Philippe Lecheminant for useful discussions and correspondence. This work has been supported in part by the Fonds National Suisse, subsidy 200020-175539, by the NCCR 51NF40-182902 "The Mathematics of Physics" (SwissMAP), and by the ERC Synergy Grant "ReNewQuantum".

## A   Perturbative expansion around $n = 0$

In this Appendix we explain how to derive the coefficients of the perturbative series $E_\ell(n)$ (for $\kappa = 2$) as an expansion around $n = 0$, rom the Bethe ansatz equations. This is a generalization of our results for the Gaudin–Yang model in [4, 5], and it is also based on the approach introduced by Volin [26, 27].

While the branch cut of the driving term makes (2.44) hard to tackle exactly, we can extract the $n \to 0$ limit by perturbing around the result of [4, 5]. Since $n$ is related to the equation only implicitly, we must first identify how to encode the correct double scaling limit in the parameters of the equation, since in Volin's method the equation is expanded in $B \to \infty$ which is equivalent to $u \to 0$. At weak coupling $n \sim 2Bu/\pi$, and this suggests the introduction of the parameter $\beta = Bu$, which is finite in the weak coupling limit. The $n, u \to 0$ limit becomes $\beta, 1/B \to 0$. The power series in $1/B$ corresponds to the weak coupling expansion in $\gamma$ as defined in (2.10) and the power series in $\beta^2$ corresponds to the power series in $n^2$.

We will work with a finite truncation of (2.44) to some finite power $\beta^{2J}$,

$$\frac{f(x)}{2} + \frac{1}{2\pi}\int_{-B}^{B}\frac{1}{1+(x-x')^2}f(x')\mathrm{d}x' = \mathrm{Re}\frac{1}{\sqrt{1-\frac{\beta^2}{B^2}(x-\frac{\mathrm{i}}{2})^2}}$$

$$\approx \sum_{i=0}^{J}(-1)^i\binom{-1/2}{i}\frac{\beta^{2i}}{B^{2i}}\mathrm{Re}(x-\mathrm{i}/2)^{2i}. \tag{A.1}$$

The method presented in [27] and used in [4, 5, 26, 28, 31] hinges on the comparison of two limits, called the *edge regime* and the *bulk regime*. For the edge regime we can follow closely the strategy of [5].

We make the substitution $x = B - z/2$ and take the limit $B \to \infty$ with $z$ finite, discarding non-analytic factors suppressed as $\mathrm{e}^{-Bz}$. We reduce (A.1) to

$$\int_0^\infty (\delta(z-z') + K(z-z'))f(z')\mathrm{d}z' = \sum_{i=0}^{2J}\alpha_i(\beta,J)\left(\frac{z}{B}\right)^i, \quad z > 0 \tag{A.2}$$

where $f(z) = f(x(z))$ and $\alpha_i(\beta,J)$ are polynomials of degree $2J$ in $\beta$ which can be easily obtained from Taylor expansion. The kernel is given by $K(z) = 2(2+z^2)^{-1}$. In this regime we can apply the Wiener-Hopf method. We extend the equation to the real line and take a Fourier transform (see [5] for details). We obtain

$$(1+\tilde{K}(\omega))\mathcal{F}_+(\omega) = \sum_{k=0}^{2J}\frac{(-1)^k k!\alpha_k(\beta,J)}{\mathrm{i}^{1+k}B^k}\left(\frac{1}{(\omega-\mathrm{i}\epsilon)^{k+1}} - \frac{1}{(\omega+\mathrm{i}\epsilon)^{k+1}}\right) + h_-(\omega), \tag{A.3}$$

where $\tilde{K}(\omega)$ is the Fourier transform of the kernel and $h_-(\omega)$ is an unknown function. The solution to this problem is

$$\mathcal{F}_+(\omega) = -G_-(0)G_+(\omega)\left(\frac{2\sum_{i=0}^{J}(-1)^i\binom{-1/2}{i}\beta^{2i}}{\mathrm{i}\omega} + \frac{1}{Bs}\sum_{m=0}^{\infty}\sum_{n=0}^{m+1}\frac{Q_{n,m-n+1}}{B^m(\mathrm{i}\omega)^n}\right), \tag{A.4}$$

where

$$G_+(\omega) = \frac{\mathrm{e}^{-\frac{\omega}{i\pi}\left(\log\left(\frac{\omega}{i\pi}\right)-1\right)}}{\sqrt{2\pi}}\Gamma\left(\frac{\omega}{i\pi}+\frac{1}{2}\right) \tag{A.5}$$

is given by the standard Wiener–Hopf decomposition of the kernel,

$$1+\tilde{K}(\omega) = \frac{1}{G_+(\omega)G_-(\omega)}, \qquad G_-(\omega) = G_+(-\omega), \tag{A.6}$$

and the coefficients $Q_{n,k}$ are *a priori* unknown, since the Wiener-Hopf method does not have enough information to fix them.

To study the bulk regime it is useful to introduce the resolvent

$$R(x) = \int_{-B}^{B}\frac{f(x')\mathrm{d}x'}{x-x'} = \sum_{k=1}^{\infty}\frac{1}{x^{k+1}}\int_{-B}^{B}x'^k f(x')\mathrm{d}x', \tag{A.7}$$

and define

$$D = \mathrm{e}^{\frac{\mathrm{i}}{2}\partial_x}, \quad R^\pm(x) = R(x\pm\mathrm{i}\epsilon) \quad \text{s.t.} \quad R^+(x) - R^-(x) = -2\pi\mathrm{i}f(x). \tag{A.8}$$

We can then mold (2.44) into

$$(1 + D^2)R^+(x) - (1 + D^{-2})R^-(x) = -(D + D^{-1})\frac{2\pi\mathrm{i}}{\sqrt{1 - x^2\beta^2/B^2}}. \tag{A.9}$$

The bulk limit is given taking $B \to \infty$ while keeping $y = x/B$ finite. In this limit, we can rearrange (A.9) into

$$R^+(x) - R^-(x) = -\frac{D - D^-}{D + D^-}(R^+(x) + R^-(x)) - \frac{2}{D + D^-}\left(\frac{2\pi\mathrm{i}}{\sqrt{1 - x^2\beta^2/B^2}}\right). \tag{A.10}$$

To work order by order in $1/B$, we organize the resolvent into

$$R(By) = R_0(y) + \frac{1}{B}R_1(y) + \frac{1}{B^2}R_2(y) + \cdots \tag{A.11}$$

$$\delta R_i(y) = R_i^+(y) - R_i^-(y), \quad y \in (-1, 1), \tag{A.12}$$

$$\Sigma R_i(y) = R_i^+(y) + R_i^-(y), \quad y \in (-1, 1). \tag{A.13}$$

By expanding the operators $D \approx 1 + \frac{\mathrm{i}}{2B}\partial_y + \cdots$ in (A.10) we can write at each order

$$\mathrm{i}\delta R_m(y) = \sum_{k=0}^{\lfloor\frac{m-1}{2}\rfloor} \frac{2(-1)^k\left(4^{k+1} - 1\right)B_{2k+2}}{\Gamma(2k+3)}\partial_y^{2k+1}\Sigma R_{m-2k-1}(y)$$
$$+ 2^{1-m}\pi E_m \sum_{k=\frac{m}{2}}^{J}(-1)^{k-\frac{m}{2}}\binom{-\frac{1}{2}}{k}\binom{2k}{m}\beta^{2k}y^{2k-m}, \tag{A.14}$$

where $B_k$ are the Bernoulli numbers and $E_m$ are the Euler numbers (which are zero for odd $m$). These equations can be solved for the discontinuous part of the resolvent order by order. However the resolvent can also have *a priori* a continuous part which contributes to $\Sigma R_i$ but not to $\delta R_i$.

Nevertheless, at this stage one can identify an adequate ansatz. At a given truncation $J$, we have

$$R(By) = \sum_{m=0}^{\infty} \sum_{n=\min[1,-\lfloor J-m/2\rfloor]}^{m} \sum_{k=0}^{m+1} c_{n,m-n,k}\frac{y^{1-k \bmod 2}}{B^m(y^2-1)^n}\left[\log\left(\frac{y-1}{y+1}\right)\right]^k. \tag{A.15}$$

This is a modified version of what was used for the Gaudin–Yang model in [4, 5]. To find the coefficients $c_{n,m-n,k}$ we need to combine three strategies.

First and foremost, one of the key properties that fixes the resolvent is the large $x$ behavior $R(x) \sim \mathrm{const}/x + \mathcal{O}(1/x^2)$. Thus we can fix each coefficient $c_{n\leq0,m-n,0}$ (i.e. the holomorphic part of the resolvent) as linear combinations of the remaining $c_{n\leq0,m-n,k\geq1}$ by setting to zero the corresponding term of order $y^{1-2n}B^m$ in the expansion $y \to \infty$. Concretely we find

$$c_{n\leq0,m-n,0} = -\sum_{j=-J}^{n}\sum_{k=1}^{m+1} I_{-n,j,k}c_{j,m-j,k}, \tag{A.16}$$

where $I_{n,m,k}$ are defined through

$$\frac{y^{1-k \bmod 2}}{(y^2-1)^n}\left[\log\left(\frac{y-1}{y+1}\right)\right]^k = \sum_{m=0}^{-n} I_{m,n,k}y(y^2-1)^m + \mathcal{O}\left(\frac{1}{y}\right). \tag{A.17}$$

As the second step we have the key insight of [26], that at each order in the $1/B$ expansion, the resolvent connects to the edge analysis through

$$\int_{-i\infty}^{i\infty}\frac{\mathrm{d}z}{2\pi\mathrm{i}}\mathrm{e}^{-sz}R(B-z/2) = \mathcal{F}_+(\mathrm{i}s) + \mathcal{O}(\mathrm{e}^{-Bs}), \tag{A.18}$$

which, when expanded in $s \to 0$ suffices to fix the coefficients $c_{n>0,m-n,k}$ and $Q_{n,m-n}$ for a given order $m$, provided all (finitely many) coefficients with $m' < m$ are known. We are left with finding $c_{n\leq 0,m-n,k\geq 1}$, for which we finally resort to (A.14).

Working through this procedure we extract the resolvent as a power series in $1/B$ and $\beta$, however we require the energy as a power series in $\gamma$ and $n$. The moments

$$\langle x^k \rangle \equiv \int_{-B}^{B} x^k f(x)\mathrm{d}x \tag{A.19}$$

can be extracted from the resolvent by using the expansion (A.7) and the bulk ansatz (A.15). This gives

$$\frac{1}{\gamma} = \frac{\langle x^0 \rangle}{\pi}, \qquad n = \frac{\beta\langle x^0 \rangle}{B\pi}. \tag{A.20}$$

These series can be inverted to express $\beta$ and $1/B$ as power series in $n$ and $\gamma$. Finally we obtain, for the ground state energy density,

$$E(n,u) = -\frac{2u}{\pi}\int_{-B}^{B}\mathrm{Re}\sqrt{1-u^2(x-\mathrm{i}/2)^2}f(x)\mathrm{d}x$$
$$\approx -\sum_{i=0}^{J}\sum_{j=0}^{i}(-1)^j\frac{2^{1-2(i-j)}}{\pi}\binom{\frac{1}{2}}{i}\binom{2i}{2j}\left(\frac{\beta}{B}\right)^{2i+1}\langle x^{2j}\rangle. \tag{A.21}$$

Let us work out an example, for $J = 1$ up to order $1/B$. The bulk ansatz (A.15) gives us

$$R(y) = y\left(y^2-1\right)c_{-1,1,0} + \left(y^2-1\right)c_{-1,1,1}\log\left(\frac{y-1}{y+1}\right) + yc_{0,0,0} + c_{0,0,1}\log\left(\frac{y-1}{y+1}\right)$$

$$+\frac{1}{B}\left(\frac{yc_{1,0,0}}{y^2-1} + \frac{yc_{1,0,2}\log^2\left(\frac{y-1}{y+1}\right)}{y^2-1} + \frac{c_{1,0,1}\log\left(\frac{y-1}{y+1}\right)}{y^2-1} + yc_{0,1,0}\right. \tag{A.22}$$

$$\left. +yc_{0,1,2}\log^2\left(\frac{y-1}{y+1}\right) + c_{0,1,1}\log\left(\frac{y-1}{y+1}\right)\right) + \mathcal{O}\left(\frac{1}{B^2}\right) + \mathcal{O}(\beta^4).$$

At infinity we have

$$R(y) \approx \left(y^3 c_{-1,1,0} + y\left(-c_{-1,1,0} - 2c_{-1,1,1} + c_{0,0,0}\right) + \mathcal{O}\left(\frac{1}{y}\right)\right)$$
$$+\frac{1}{B}\left(c_{0,1,0}y + \mathcal{O}\left(\frac{1}{y}\right)\right) + \mathcal{O}\left(\frac{1}{B^2}\right) \tag{A.23}$$

which sets

$$c_{-1,1,0} = c_{0,1,0} = 0, \quad c_{0,0,0} = 2c_{-1,1,1}. \tag{A.24}$$

For the edge-bulk matching, we have from (A.4)

$$\mathcal{F}(\mathrm{i}s) = \frac{2 + \beta^2}{2s} - \frac{\left(2 + \beta^2\right)\left(\log\left(\frac{4s}{\pi}\right) + \gamma_E - 1\right)}{2\pi} + \cdots, \tag{A.25}$$

and from the ansatz

$$\int_{-\mathrm{i}\infty}^{\mathrm{i}\infty} \frac{\mathrm{d}z}{2\pi\mathrm{i}} \mathrm{e}^{-sz} R(B - z/2) = -\frac{c_{0,0,1}}{s} + c_{1,0,0} - c_{1,0,1}(\log(4Bs) + \gamma_E)$$
$$+ \frac{1}{6}c_{1,0,2}\left(6\log(4Bs)(\log(4Bs) + 2\gamma_E) + 6\gamma_E^2 - \pi^2\right) + \cdots, \tag{A.26}$$

where $\gamma_E$ is the Euler-Mascheroni constant. Comparing (A.25) with (A.26) yields

$$c_{0,0,1} = -\frac{\beta^2}{2} - 1, \quad c_{1,0,0} = \frac{\left(\beta^2 + 2\right)\left(\log(\pi B) + 1\right)}{2\pi}, \quad c_{1,0,1} = \frac{\beta^2 + 2}{2\pi}, \quad c_{1,0,2} = 0. \tag{A.27}$$

Last but not least we have the perturbative equations from (A.14)

$$\mathrm{i}\delta R_0(y) = \left(1 + \frac{y^2\beta^2}{2}\right), \qquad \mathrm{i}\delta R_1(y) = \partial_y \Sigma R_0(y). \tag{A.28}$$

The first one leads to

$$2\pi\mathrm{i}\left(y^2 - 1\right)c_{-1,1,1} + 2\pi\mathrm{i}c_{0,0,1} = -\mathrm{i}\left(1 + \frac{y^2\beta^2}{2}\right) \Rightarrow c_{-1,1,1} = -\frac{\beta^2}{4\pi}, \tag{A.29}$$

while the second one to

$$2\pi\mathrm{i}\frac{c_{1,0,1}}{y^2 - 1} + 2\pi\mathrm{i}c_{0,1,1} + 2\pi\mathrm{i}\log\left(\frac{1 - y}{y + 1}\right)\left(\frac{2yc_{1,0,2}}{y^2 - 1} + 2yc_{0,1,2}\right) =$$
$$-\mathrm{i}\left(\frac{2c_{0,0,1}}{y^2 - 1} + 2c_{-1,1,1} + c_{0,0,0}\right) - 2\mathrm{i}yc_{-1,1,1}\log\left(\frac{1 - y}{y + 1}\right). \tag{A.30}$$

Plugging in the previous results (A.27) in these equations fixes the remaining coefficients,

$$c_{0,1,1} = \frac{\beta^2}{\pi}, \quad c_{0,1,2} = \frac{\beta^2}{4\pi}. \tag{A.31}$$

With these coefficients and (A.7) we have

$$\langle x^0 \rangle = \frac{1}{3}\left(\beta^2 + 6\right)B + \frac{-\beta^2 + \left(\beta^2 + 2\right)\log(\pi B) + 2}{2\pi} + \cdots$$
$$\langle x^2 \rangle = \frac{1}{15}\left(3\beta^2 + 10\right)B^3 + \frac{\left(\beta^2 + 2\right)\left(\log(\pi B) - 1\right)}{2\pi}B^2 + \cdots \tag{A.32}$$
$$\langle x^4 \rangle = \frac{1}{35}\left(5\beta^2 + 14\right)B^5 + \frac{\left(-13\beta^2 + 9\left(\beta^2 + 2\right)\log(\pi B) - 30\right)}{18\pi}B^4 + \cdots$$

Using the $\langle x^0 \rangle$ moment in tandem with (A.20), we can reverse the series into

$$\frac{1}{B} = \gamma \left( \frac{2}{\pi} + \frac{\pi n^2}{12} + \mathcal{O}\left(n^3\right) \right) + \gamma^2 \left( \frac{2 - \log\left(\frac{4\gamma^2}{\pi^4}\right)}{\pi^3} - \frac{\left(5 + \log\left(\frac{4\gamma^2}{\pi^4}\right)\right) n^2}{12\pi} + \mathcal{O}\left(n^3\right) \right) + \mathcal{O}\left(\gamma^3\right),$$

$$\beta = \left( \frac{\pi n}{2} - \frac{\pi^3 n^3}{48} + \mathcal{O}\left(n^4\right) \right) + \gamma \left( \frac{\left(\log\left(\frac{2\gamma}{\pi^2}\right) - 1\right) n}{2\pi} + \frac{7\pi n^3}{48} + \mathcal{O}\left(n^4\right) \right) + \mathcal{O}(\gamma^2).$$

$$(A.33)$$

Feeding (A.32) and (A.33) into (A.21), one obtains at last

$$E(n, u) = \left( -\frac{\pi^2}{12} + \frac{\pi^4 n^2}{960} + \mathcal{O}\left(n^4\right) \right) + \gamma \left( \frac{1}{2} + \mathcal{O}\left(n^4\right) \right) + \cdots \qquad (A.34)$$

This procedure can be carried out to arbitrary order in $\beta$ (or $n$) and $1/B$ (or $\gamma$), though at exponentially increasing computational cost.

Among the results found using this procedure is the expansion of the coefficient $E_2(n)$ as a Taylor series in $n$. The very first terms can be found in (3.10). This expression has an interesting number-theoretic implication. First of all, the resulting series seems to be convergent. When $n = 1$, the series should equal the value obtained in (2.30),

$$E_2(1) = \frac{7\zeta(3)}{4\pi^3}. \qquad (A.35)$$

Setting $n = 1$, reorganizing the terms and comparing to the exact result we find

$$\zeta(3) = \pi^3 \left( \frac{1}{42} + \sum_{k=1}^{\infty} a_k \zeta(2k) \right), \qquad (A.36)$$

where

$$a_1 = \frac{1}{336}, \, a_2 = \frac{1}{448}, \, a_3 = \frac{127}{86016}, \, a_4 = \frac{2099}{2064384}, \, a_5 = \frac{26937}{36700160}, \, a_6 = \frac{2800299}{5071962112}, \, \cdots \quad (A.37)$$

Up to $k = 16$ we find the $a_k$ to be rational numbers. This is of note since most representations of the Apery constant $\zeta(3)$ which involve $\zeta(2k)$ and rational coefficients have an overall factor of $\pi^2$ [66], while this result implies the existence of a representation with an overall factor of $\pi^3$.

## B  Another derivation of the trans-series

In [11], Misurkin and Ovchinnikov represented the function $\mathcal{I}(z)$ defined in (2.49) as

$$\mathcal{I}(z) = \frac{1}{\pi} - \frac{z}{8} + \frac{z^2}{2\pi^4} \int_1^{\infty} \frac{\mathrm{d}y \sqrt{y^2 - 1}}{y^3} \int_0^{2\pi y/z} \frac{x^2 \mathrm{d}x}{\sinh(x)\sqrt{1 - \left(\frac{xz}{2\pi y}\right)^2}}. \qquad (B.1)$$

The perturbative series (2.30) was obtained in [11] as the asymptotic expansion of the integral in (B.1), after extending the integration domain in the second integral from $2\pi y/z$ to infinity.

Therefore, the trans-series should be given by the asymptotic expansion of the exponentially small error made in performing this extension. This error is given by $\pm i$, times the integral

$$\frac{z^2}{2\pi^4}\int_1^\infty \frac{\mathrm{d}y\sqrt{y^2-1}}{y^3}\int_{2\pi y/z}^\infty \frac{x^2\mathrm{d}x}{\sinh(x)\sqrt{\left(\frac{xz}{2\pi y}\right)^2-1}}. \tag{B.2}$$

The $\pm i$ is due to the ambiguity in extracting the square root and turns out to correspond to the two possibilities for lateral Borel resummation. Let us denote

$$\mathcal{F}(x) = \frac{x^2}{\sinh(x)\sqrt{\left(\frac{xz}{2\pi y}\right)^2-1}}. \tag{B.3}$$

Then, by simply shifting the integration variable, we have

$$\int_{2\pi y/z}^\infty \mathcal{F}(x)\mathrm{d}x = \int_0^\infty \mathcal{F}\left(\frac{2\pi y}{z}+w\right)\mathrm{d}w. \tag{B.4}$$

The integrand includes

$$\frac{1}{\sinh\left(\frac{2\pi y}{z}+w\right)} = 2\sum_{r=0}^\infty \exp\left\{-(2r+1)\left(\frac{2\pi y}{z}+w\right)\right\}. \tag{B.5}$$

Let us focus on the first term $r=0$. We write the integral over $w$ in (B.2) as

$$\sqrt{2}\left(\frac{2\pi y}{z}\right)^{5/2}\int_0^\infty \mathrm{e}^{-w}w^{1/2}\left(1+\frac{wz}{2\pi y}\right)^2\left(1+\frac{wz}{4\pi y}\right)^{-1/2}\mathrm{d}w. \tag{B.6}$$

We now define the coefficients $a_n$ as

$$(1+x)^2\left(1+\frac{x}{2}\right)^{-1/2} = \sum_{n\geq 0} a_n x^n. \tag{B.7}$$

The integral (B.6) has then the asymptotic expansion

$$\sqrt{2}\sum_{n\geq 0} a_n\Gamma\left(n+1/2\right)\left(\frac{z}{2\pi y}\right)^{n-5/2}. \tag{B.8}$$

After an appropriate change of variables, we can calculate the integral over $y$ as

$$\int_1^\infty \mathrm{e}^{-2\pi y/z}\sqrt{y^2-1}y^{-n-1/2}\mathrm{d}y = \mathrm{e}^{-2\pi/z}\sqrt{2}\left(\frac{z}{2\pi}\right)^{3/2}j_n(z), \tag{B.9}$$

where $j_n(z)$ have the asymptotic expansion

$$j_n(z) \sim \sum_{m\geq 0} d_{n,m}\Gamma\left(m+3/2\right)\left(\frac{z}{2\pi}\right)^m, \tag{B.10}$$

and the coefficients $d_{n,m}$ are defined as

$$\sqrt{1+x/2}\,(1+x)^{-n-1/2} = \sum_{m\geq 0} d_{n,m}x^m. \tag{B.11}$$

Putting everything together, we find that (B.2) leads to the following trans-series for the energy:

$$\frac{1}{2} z \, \mathrm{e}^{-2\pi/z} \varphi_1(z) + \mathcal{O}(\mathrm{e}^{-6\pi/z}), \tag{B.12}$$

where

$$\varphi_1(z) = \frac{16}{\pi^3} \sum_{n,m=0}^{\infty} a_n d_{n,m} \Gamma(n+1/2) \Gamma(m+3/2) \left(\frac{z}{2\pi}\right)^{n+m}. \tag{B.13}$$

We have checked by explicit evaluation of the first terms that this agrees with (4.49). The factor $1/2$ in (B.12) is natural since here we are computing the formal imaginary part of the trans-series, which is half of the discontinuity. It is also easy to see that the terms with $r > 0$ in (B.5) reproduce precisely the terms with $r > 0$ in the r.h.s. of (4.52).

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
