# Peer review of "Resurgence and renormalons in the one-dimensional Hubbard model"

_SciPost Physics_

## Round 2 · Referee Report · Anonymous (Referee 1) · 2021-10-5

Strengths

This paper studies the one-dimensional Hubbard model and uses different approaches to study the perturbative and non-perturbative contributions to its ground state energy. The main strengths of the paper can be summarised as:

  1. The relation between the non-perturbative contributions and the energy gap due to renormalons: in the two-component case the use of integrability and resurgence/trans-series techniques allowed the prediction of the non-perturbative contribution to the ground state energy, associated to its divergent formal perturbative expansion. This non-perturbative exponential and more its prefactor were then obtained by studying the zero and the half-filled limiting cases. The authors then show that these results are consistent with the existence of certain renormalon diagrams.

  2. A conjecture for the full ground state energy in the multi-component case for arbitrary filling fractions, by using all the results from different limiting cases.

  3. Using tools of resurgence and trans-series to determine renormalon ring diagrams contributing non-perturbatively: this analysis is particularly interesting and elucidating at the half-filling limit as many results can be obtained exactly.

Weaknesses

The paper is generally well written and well presented, well referenced and detailed, and without any major weaknesses. The only issue I would say this paper currently has is:

  1. In section 2, the authors discuss several different limits/cases of the multi-component Hubbard model, but they are not clearly separated. It would definitely help the reader if the authors distinguished them more clearly, or summarized them in a paragraph at the beginning.

Report

This paper presents novel theoretical results and conjectures as well as their physical interpretation for the one dimensional Hubbard model. It also shows the strengths of different methods in addressing this problem and how to use them effectively. It is my opinion that this paper meets the expectations and all the acceptance criteria, and it should be published in SciPost after some minor revision.

Requested changes

There are a few minor changes that I would suggest the authors should take into consideration:

  1. as mentioned above, in section 2, the authors discuss several different limits/cases of the multi-component Hubbard model, but they are not clearly separated. It would definitely help the reader if the authors distinguished them more clearly, or summarized them in a paragraph at the beginning.

  2. In section 2.1 a general reference to the Hubbard model and its main features is missing

  3. Equation (2.6) is a different representation of (2.4), so it would be useful to add H_I= also in the latter.

  4. Just after equation (2.20) the authors state that the Fermi momentum is also kept fixed. How is this achieved given the definition (2.8)?

  5. Can the authors provide a reference for equation (2.23)?

  6. Just after figure 2, when the coefficient E_2(n,kappa) is mentioned, it would be useful to recall where it was defined (equation (2.10) or (2.17)).

  7. In section 2.3 the parameter B first appearing in equation (2.44) was never defined.

  8. In section 3.2 it is mentioned that other models have been studied using Volin's method. The authors should add references to some of these papers.

  9. equation (4.5) looks like a single parameter trans-series, although the authors have added multiple parameters. Usually one would have c^ell instead of c_ell for this shape of a trans-series, why is this different?

  10. Just after equation (4.13), what is RPA?

  11. equation (4.66) is somewhat misleading. The median summation has this form because it fixes the trans-series parameters such as the lateral summations of the full trans-series match. The authors should add a note to this effect. In the current case these lateral summations are very simple and this cancellation must happen automatically, is that the case?

---

## Round 3 · Referee Report · Anonymous (Referee 2) · 2022-9-5

Strengths
A recent conjecture proposed by the same authors for Fermi systems with attractive interactions relates the gap in the spectrum, which is non-perturbative in the coupling constant, to the large order behaviour of the perturbative expansion.
For the two component Hubbard model the authors use integrability to obtain the perturbative expansion of the ground state energy, and verify their conjecture connecting the large order behaviour of this expansion to the energy gap.
Very interestingly the authors are able to predict the asymptotic gap for the non-integrable multicomponent case using resurgence analysis.
Weaknesses
Report
This paper meets all the criteria to be published in SciPost after some very minor revisions.
Requested changes
I have only two minor suggestions for the authors.
1) Hartree Fock and Ring diagrams are mentioned in section 2.1 and sketched in Figure1. Perhaps the authors can elaborate a bit more on this, for example adding to the caption an explanation of the notation for dashed vs continuous lines in the diagrams.
2)The authors should clarify the relation between E^ring_l(n,k) in eq. 2.23 and E_l(n,k) in 2.14. What else can contribute to E_l(n,k)?

Tomas Reis on 2022-01-07 [id 2076]
We would like to thank the referee for their attentive reading of our manuscript. We have incorporated all the suggestions in this revision of the paper, and included the requested clarifications.
Perhaps we can comment more particularly on point (9) raised by the referee. In trans-series arising from ordinary differential equations, the trans-series parameters are all related, and for equations of first order, the coefficient of the \ell-th “instanton" correction is indeed of the form C^\ell. However, in trans-series appearing in QFT, we do not have any reason to believe that this will be also the case (or at least we are not aware of any result going into that direction). That’s why we have decided to write a more generic trans-series form.
The point (11) has been also addressed. It turns out that all the series appearing in the trans-series (except the perturbative one) are Borel summable along the positive real axis, so the median resummation is much simplified.

---

## Editorial Decision

unknown